# Structure of a nucleosome-bound MuvB transcription factor complex reveals DNA remodelling

Marios G. Koliopoulos[1,3], Reyhan Muhammad[1,3], Theodoros I. Roumeliotis[2], Fabienne Beuron [1], Jyoti S. Choudhary [2] & Claudio Alfieri[1] ✉

Genes encoding the core cell cycle machinery are transcriptionally regulated by the MuvB family of protein complexes in a cell cycle-specific manner. Complexes of MuvB with the transcription factors B-MYB and FOXM1 activate mitotic genes during cell proliferation. The mechanisms of transcriptional regulation by these complexes are still poorly characterised. Here, we combine biochemical analysis and in vitro reconstitution, with structural analysis by cryo-electron microscopy and cross-linking mass spectrometry, to functionally examine these complexes. We find that the MuvB:B-MYB complex binds and remodels nucleosomes, thereby exposing nucleosomal DNA. This remodelling activity is supported by B-MYB which directly binds the remodelled DNA. Given the remodelling activity on the nucleosome, we propose that the MuvB:B-MYB complex functions as a pioneer transcription factor complex. In this work, we rationalise prior biochemical and cellular studies and provide a molecular framework of interactions on a protein complex that is key for cell cycle regulation.

During the cell cycle, a cell replicates its genome into two identical copies and coordinates the chromosome segregation with cell division. The orderly progression through the different phases of the cell cycle relies on the cyclin-dependent kinase-cyclin (CDK-cyclin) oscillator that sequentially triggers cell cycle transitions by phosphorylation of specific targets including the downstream effectors of the cell cycle. The oscillating activity of the CDK-cyclins is achieved by cell cycle specific protein ubiquitination, phosphorylation and transcriptional regulation[1,2].

Cell cycle-dependent transcription of cell cycle genes is controlled by the MuvB family of transcriptional regulators[3–5]. The MuvB core complex consists of a main scaffolding subunit called LIN9, the histone-binding protein RbBP4, the DNA-binding protein LIN54, and the smaller subunits LIN37 and LIN52[6]. The DNA-binding domain of LIN54 recruits MuvB to its target genes at a specific sequence termed the cell cycle homology region (CHR)[7,8]. MuvB target promoters are TATA-less and

the CHR is located directly upstream of the transcription start site (TSS) and the +1 nucleosome[8,9]. The presence of the +1 nucleosome generates a barrier for the assembly of the basal transcriptional apparatus[10], and its positioning and dynamics are finely regulated to define the transcriptional status of each gene. Even though the mechanisms of regulation of the +1 nucleosome are not fully understood and are the object of intensive research[11], the location of the CHR and several studies[7,9,12] suggests that MuvB complexes, which perform both activating and repressive roles, may be involved in this process.

During cell cycle exit in quiescence and in senescence, also known as G0, the MuvB complex interacts with a retinoblastoma-like protein, thereby forming the DREAM complex[3,4,13], which mediates transcriptional repression of about 1000 cell cycle genes[6]. Recognition of the target genes by the DREAM complex is combinatorial, as it involves binding of a CHR and an upstream cell-cycle dependent element (CDE)[2,14,15].

[1]Division of Structural Biology, Chester Beatty Laboratories, The Institute of Cancer Research, London, UK. [2]Functional Proteomics, Chester Beatty Laboratories, Cancer Biology Division, The Institute of Cancer Research, London, UK. [3]These authors contributed equally: Marios G. Koliopoulos, Reyhan Muhammad. ✉e-mail: claudio.alfieri@icr.ac.uk

When the cell is committed to divide, the increasing activity of CDK2-cylin D and E at the G1-S transition promotes the hyperphosphorylation of the retionoblastoma-like proteins[16,17], thereby causing the disassembly of DREAM and cell cycle re-entry[16,18,19]. The remaining CHR-bound MuvB core complex sequentially associates with B-MYB and FOXM1 transcription factors (TFs) during S phase[20]. B-MYB and FOXM1 are oncogenes and are overexpressed in several cancer types[21–24]. Sequential assembly of B-MYB and FOXM1 onto the MuvB complex at CHR elements is required for triggering transcriptional activation of the mitotic gene programme at the G2 phase[20,25,26]. Importantly, the complex of B-MYB and MuvB (MMB) is required for FOXM1 recruitment at cell cycle genes, therefore B-MYB has been defined as a pioneer TF for cell cycle-dependent transcription of G2/M genes[20,27].

The molecular mechanism for MuvB-dependent gene repression or activation is poorly understood, and the stoichiometry and overall architecture of this complex are unclear.

The presence of the histone-binding protein RbBP4, which is shared among several chromatin regulator complexes[2,28–30], suggests that this complex is involved in the regulation of chromatin structure, but the lack of any ATPase domain in MuvB indicates that it is not involved in ATP-dependent chromatin remodelling. Many chromatin regulating complexes function in an ATP-independent fashion. For example, Polycomb repressive complex 1 (PRC1) utilises oligomerisation as part of its inhibitory mechanism[31–33]; pioneer transcription factors establish competence for gene expression by binding the nucleosomal DNA and initiating chromatin remodelling thereby exposing additional nucleosomal DNA to downstream TFs and to the transcriptional apparatus[34–38]. Histone chaperones assist nucleosome assembly and disassembly during transcription, DNA replication, and repair[39,40]. It is unknown if the MuvB complex could utilise any of the mechanisms listed above for actuating its transcriptional regulating functions.

Here, we set out to investigate whether MuvB uses one of these known ATP-independent chromatin remodelling mechanisms or possibly a previously undescribed mechanism. Therefore, we determined a structural snapshot of a MuvB complex in action using cryo-electron microscopy (cryo-EM), in the process of remodelling a nucleosome, and define its stoichiometry, assembly, and chromatin binding. Taken together, our results suggest that MMB is a pioneer transcription factor, and we derive a model according to which its nucleosome remodelling activity could be contributing to transcriptional activation in the context of cell proliferation.

## Results

### Biochemical characterisation of MuvB:nucleosome complexes

Given the compelling evidence that the MuvB complex is involved in nucleosome binding near the CHR-containing promoter of target genes[7,9,12], we decided to attempt reconstitution of several MuvB complexes with the nucleosome. We recombinantly expressed all the subunits of the MuvB complex including LIN9, LIN37, LIN52, RbBP4, and a short version of LIN54 (LIN54sh) where the non-conserved N-terminus was removed (Fig. 1a) in a baculovirus/insect cell expression system. This resulted in the preparation of a highly pure protein complex that eluted in a sharp and symmetric peak in size exclusion chromatography (SEC) (Supplementary Fig. 1a).

To analyse the interaction between MuvB and the nucleosome specifically and independently from the MuvB CHR binding function, we reconstituted a MuvB complex lacking LIN54, which is required for recognising the CHR sequence. We named this MuvB subcomplex core MuvB (MuvB[core]). MuvB[core] is as stable as the full MuvB complex (Supplementary Figs. 1a, b and 2a, b), demonstrating that the absence of LIN54 does not perturb the structural integrity of MuvB. We, therefore, used this complex to probe its interaction with a model nucleosome reconstituted with a 167 base pairs (bp) DNA containing

the 601 Widom sequence by electrophoretic mobility shift assays (EMSA). The MuvB[core] complex binds the nucleosome when present at 4-fold molar excess (Supplementary Fig. 2e). Expectedly, this complex binds a CHR-containing nucleosome with a lower affinity than the LIN54sh-containing MuvB complex (Supplementary Fig. 2g). Importantly, the affinity of MuvB[core] for a nucleosome is higher than for free DNA (Supplementary Fig. 3a, b).

In order to define the subunits that directly interact with the nucleosome and the binding mode of the MuvB complex in respect of the nucleosome, we attempted reconstitution of a MuvB[core]:nucleosome complex suitable for structural analysis by cryo-EM. The nucleosome complex with MuvB[core]:B-MYB (MMB[core]) (Supplementary Fig. 2f, lanes 6–9) manifested better homogeneity than nucleosome complexes with MuvB[core] (Supplementary Fig. 2e, lanes 1–5) and with DREAM[core] (Supplementary Fig. 2f, lanes 1–5), as shown by EMSA (Supplementary Fig. 2e, f). In fact MMB[core]:nucleosome shifted band is sharper and more defined, unlike the other complexes tested (Supplementary Fig. 2e, f).

### Cryo-EM structure of the MMB[core] complex bound to a nucleosome

To determine the structure of the MMB[core]:nucleosome complex (Fig. 1a), we used the GraFix protocol[41] to minimise disassembly during vitrification (Supplementary Fig. 3c), and collected a large dataset on a Titan Krios electron microscope equipped with an electron-counting direct detector (see "Methods", Supplementary Fig. 4a). Extensive two- and three-dimensional classification allowed us to resolve the overall structure of the highly heterogeneous and flexible MMB[core]:nucleosome complex (see "Methods", Supplementary Figs. 4, 5, and Supplementary Table 1). Our structure shows that a portion of MMB[core], which we call the MuvB[TAIL], contacts the histone octamer and is embedded in the nucleosome disc, while a flexibly attached module that we refer to as the MuvB[HEAD] reaches across the nucleosomal entry DNA and sits on the outside of the DNA gyres (Fig. 1b, c). Due to the flexibility of the complex, we employed focused refinements to resolve the structures of MuvB[HEAD] and the MuvB[TAIL] at sub-nanometre resolution and the structure of nucleosome in the context of the MMB[core]:nucleosome complex at 3.3 Å resolution (Supplementary Movie 1, Supplementary Fig. 5, Supplementary Table 1 and "Methods").

By combining our cryo-EM reconstructions of the MMB[core]:nucleosome complex with chemical crosslinking-mass spectrometry (XL-MS) data and an additional high-resolution cryo-EM structure of the MuvB apo-complex that we obtained (see below and Supplementary Figs. 6–9, Supplementary Table 1, and Supplementary Data 1), we were able to define the overall architecture of a MuvB core complex with the nucleosome (Fig. 1). Strikingly, our structure reveals that the nucleosome in the MMB[core]:nucleosome complex is remodelled. Approximately 20 base pairs of the nucleosomal DNA are bent away from their unperturbed conformation and held between the MuvB[HEAD] and MuvB[TAIL] modules (Fig. 1b, c).

### MuvB[HEAD] structure and assembly

Having identified the MuvB[HEAD] and MuvB[TAIL] modules in our reconstruction of the MMB[core]:nucleosome complex, we sought to determine the detailed structure of MuvB to facilitate a mechanistic interpretation of the nucleosome-bound complex. Mass photometry measurements of the purified MuvB complex shows a main peak at around 176 kDa (Supplementary Fig. 1b), suggesting that the five MuvB subunits are present in a 1:1:1:1:1 stoichiometry in this complex. Interestingly a minor portion of the complex shows a higher molecular weight, suggesting that this complex could be forming oligomers (Supplementary Fig. 1b). In order to investigate this further, we performed SEC coupled with multi angle light scattering (SEC-MALS) on this complex at different concentrations. This analysis shows

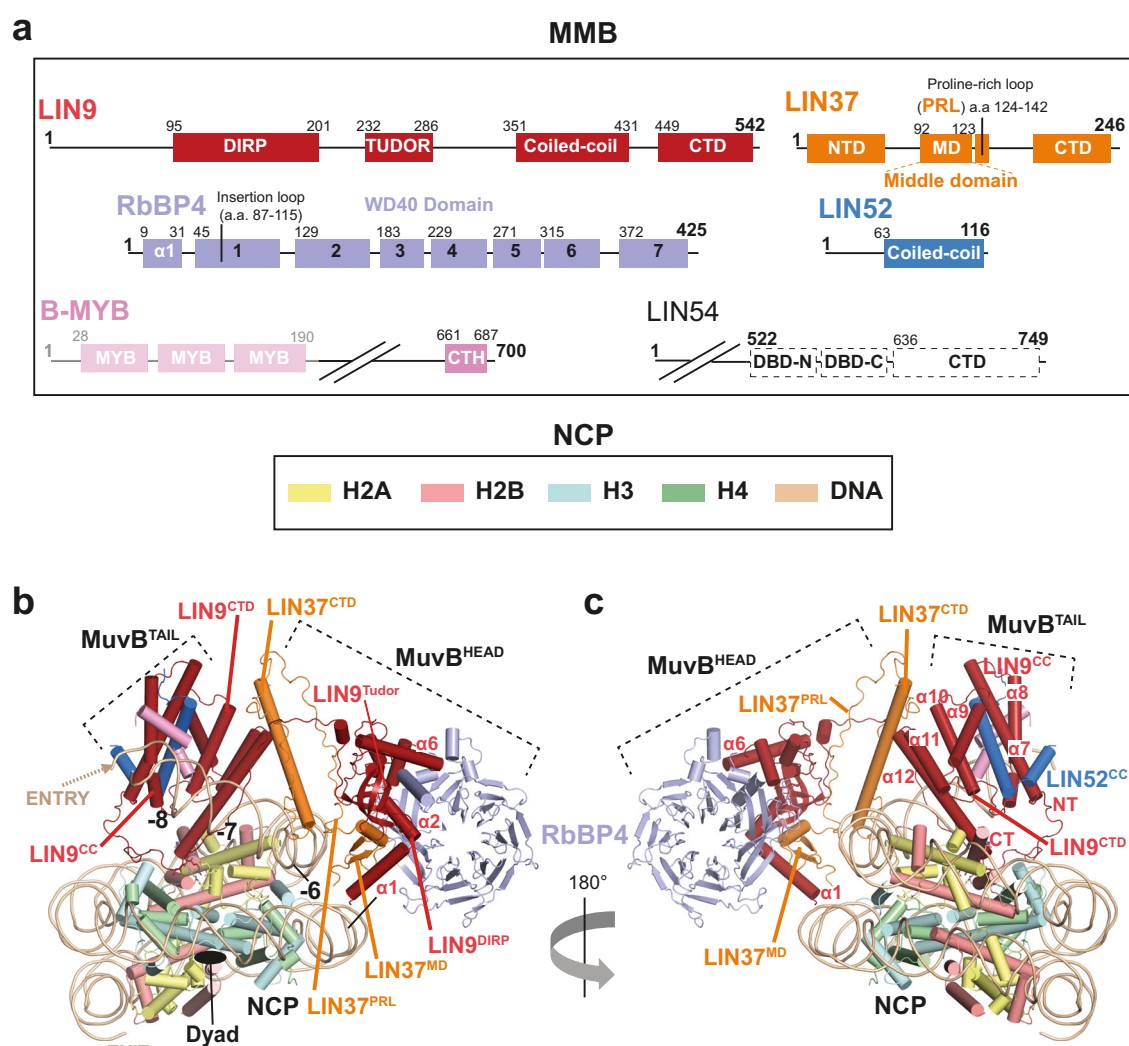

**Fig. 1 | Structure of the MMB complex with the nucleosome core particle (NCP), overall complex assembly, and binding mode. a** Schematics showing domains and functional regions on the subunits of the MMB complex represented in linear form. B-MYB N-terminus containing a DNA binding domain is disordered in the cryo-EM structure and it is represented with fading colours. LIN54 is represented with dashed lines for clarifying that this protein was not included in the core MMB (MMB^core):nucleosome structure. NCP subunit colour scheme used in this Fig. is indicated for each histone and DNA. **b, c** Cartoon representations of two main views of the MMB^core:nucleosome complex showing the overall architecture of the complex. Superhelical (SHL) positions in respect of the nucleosome dyad (0) are labelled in (**b**). DNA entry and exit is also indicated. MMB^core binding spans over three SHL positions (−8 to −6).

that MuvB can oligomerise in a concentration-dependent manner (Supplementary Fig. 1c). MuvB oligomers can be stabilised by briefly incubating the complex with glutaraldehyde causing them to elute as partially overlapping peaks in SEC (Supplementary Fig. 1a). A monomeric MuvB complex can be isolated from oligomers by SEC (Supplementary Fig. 1a). This cross-linking procedure was instrumental in obtaining a highly homogeneous sample that could be used for preparing cryo-EM grids (Supplementary Fig. 7a). We, therefore, obtained a 3D reconstruction map at 3.5 Å resolution (Supplementary Figs. 7–9). The excellent quality of the cryo-EM map (Supplementary Fig. 8 and 9) allowed the ab initio building of an atomic model of a ~70 kDa structure accounting for the WD40 domain-containing RbBP4, the Domain in Rb-related Pathway (DIRP), and tudor domain of LIN9, the middle domain (MD) and a proline-rich loop (PRL) of LIN37 (Figs. 1a–c and 2a–d). We name this portion of MuvB as RbBP4 subcomplex (Fig. 2a). Strikingly, the rest of the MuvB complex is not visible in our map indicating that these regions of the complex are highly mobile in respect of the RbBP4 subcomplex.

Except for the LIN37 PRL, we note that our results obtained using the full MuvB apo complex are consistent with the results of a recent crystallographic study reporting the structure of an isolated LIN9^DIRP-Tudor-RbBP4-LIN37^MD subcomplex[9]. The RMSD between our cryo-EM structure and the published crystal structure is of 0.658 Å.

In our structure, LIN9 amino-terminal region, including the DIRP and the tudor domain interact extensively with RbBP4 in four different sites around the conserved RbBP4 amino-terminal helix α1 (Fig. 2b–d), which is a hub of protein-protein interaction in several other RbBP4-containing complexes[28–30]. This interaction buries a total surface area of 2809 Å².

RbBP4 WD40 domain consists of a propeller made of seven blades and it exhibits a so called "top" side, which is less wide than the bottom one (Fig. 2b). At the top side, blades one and seven form a first patch for LIN9 α1 and 2, which are oriented about 90 degrees in respect to each other (Figs. 1b, 2b–d and Supplementary Fig. 9b). Furthermore, LIN9 α2 packs against the RbBP4 amino-terminal helix (Figs. 1b, 2b–d and Supplementary Fig. 10c). LIN9 α3, 4 and 5 augment additional interactions with the carboxy-terminal part of the RbBP4 amino-terminal helix, where LIN9 reaches the bottom side of the WD40 domain at blade 6 (Fig. 2c, d). The basic loop following helix 5 occupies the histone H4 binding site of RbBP4[42], by being

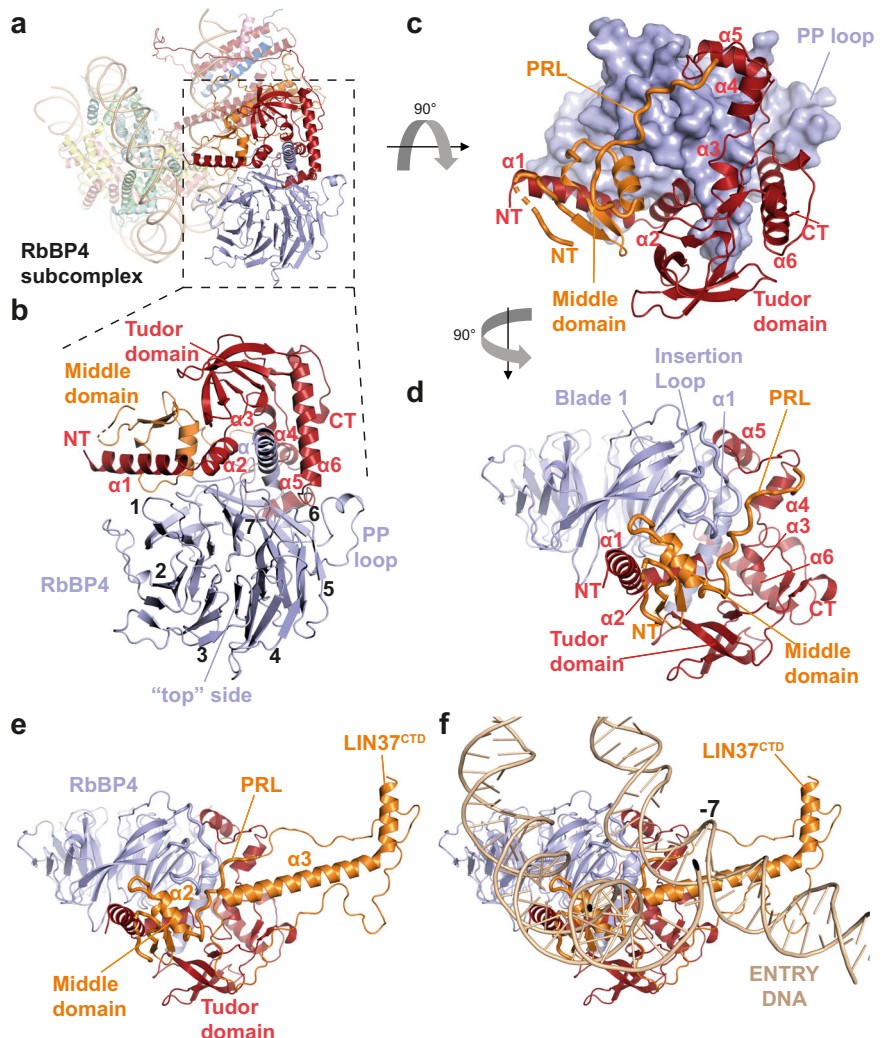

**Fig. 2 | Architecture and assembly of the MuvB^HEAD complex. a–d** Cartoon representation showing the structure of the RbBP4 subcomplex. The position of this complex in relation to the entire model is illustrated in (**a**, **b**). Blades of the RbBP4 WD40 domain are numbered in (**b**). RbBP4 is shown is surface representation in **c** to highlight the extended interaction of LIN9 N-terminal domains (DIRP and tudor) and LIN37 middle domain proline-rich loop (PRL). **e**, **f** LIN37^CTD bridges the RbBP4 complex to the nucleosomal DNA (SHL −7).

sandwiched between RbBP4 α1 and the PP loop coming from blade 6 (Fig. 2b,c). Consistently, a reconstituted MuvB complex is not able to bind H4 peptides[9].

The LIN9 tudor domain is sandwiched between LIN9 α3 and 6, and the beginning of RbBP4 α1 (Figs. 1b, 2b–d and Supplementary Fig. 9e). The latter interaction is mediated by RbBP4 Arg15 and Glu14, which interacts with the Tyr269-Glu270, Arg229-Arg231 pairs coming from LIN9 tudor domain (Supplementary Fig. 9e). Similar arginine pairs in tandem tudor domains are involved in DNA binding[43]. Given that these arginine residues are heavily interacting with RbBP4, it is unlikely that LIN9 tudor domain is involved in DNA binding. Moreover, this domain is not involved in binding of histone peptides[9], thereby suggesting that LIN9 tudor domain has exclusively a structural role in assembling the RbBP4 subcomplex and not a recruiting role to either DNA or histones. Consistently, in our MMB^core:nucleosome structure, the tudor domain is located far away from both histones or DNA and does not make any obvious contact with them (Fig. 1b).

LIN37^MD is formed by 1 helix bearing a conserved PLYxCRxW sequence (Figs. 1b, 2a–c and Supplementary Figs. 9a–c, 10a). These residues interact with LIN9 α1 and 2, and with RbBP4 insertion loop, coming from blade 1 (Supplementary Fig. 9b, c). This is followed by a short loop and a small β-sheet, which are saddled onto LIN9 α1

(Supplementary Fig. 9c). This interaction involves LIN37 Glu111 and LIN9 Arg108 and extended hydrophobic contacts (Supplementary Fig. 9c).

Strikingly, a C-terminal PRL connects LIN37 to LIN9 α4 thereby buttressing LIN9 into RbBP4 (Supplementary Fig. 9a, and Figs. 1a, c, 2c, d). Our structure suggests that LIN37 has a key role in stabilising the MuvB complex and, consistently, MuvB reconstituted without LIN37 can still form but it is less stable as shown by mass photometry (Supplementary Fig. 2c, d). This is consistent with other findings showing that LIN37 is not required for MuvB assembly[44], however, our structural-functional analysis shows in addition that it is indeed involved in complex stability.

Our RbBP4 subcomplex structure readily fits onto the MuvB^HEAD module of the MMB^core cryo-EM map where it occupies most of this cryo-EM density (Supplementary Fig. 5a,c, d and Supplementary Movie 1). The docking is unambiguous as secondary structures visible in the cryo-EM map match with the atomic coordinates (Supplementary Fig. 5d and Supplementary Movie 1). Additional density extends from the PRL of LIN37 and could be assigned unambiguously to the conserved LIN37^CTD, which folds into a kinked α-helix (Fig. 1b, c, Supplementary Figs. 5a, d, 10b, and Supplementary Movie 1). This assignment is consistent with our XL-MS data (Supplementary Fig. 6c) where

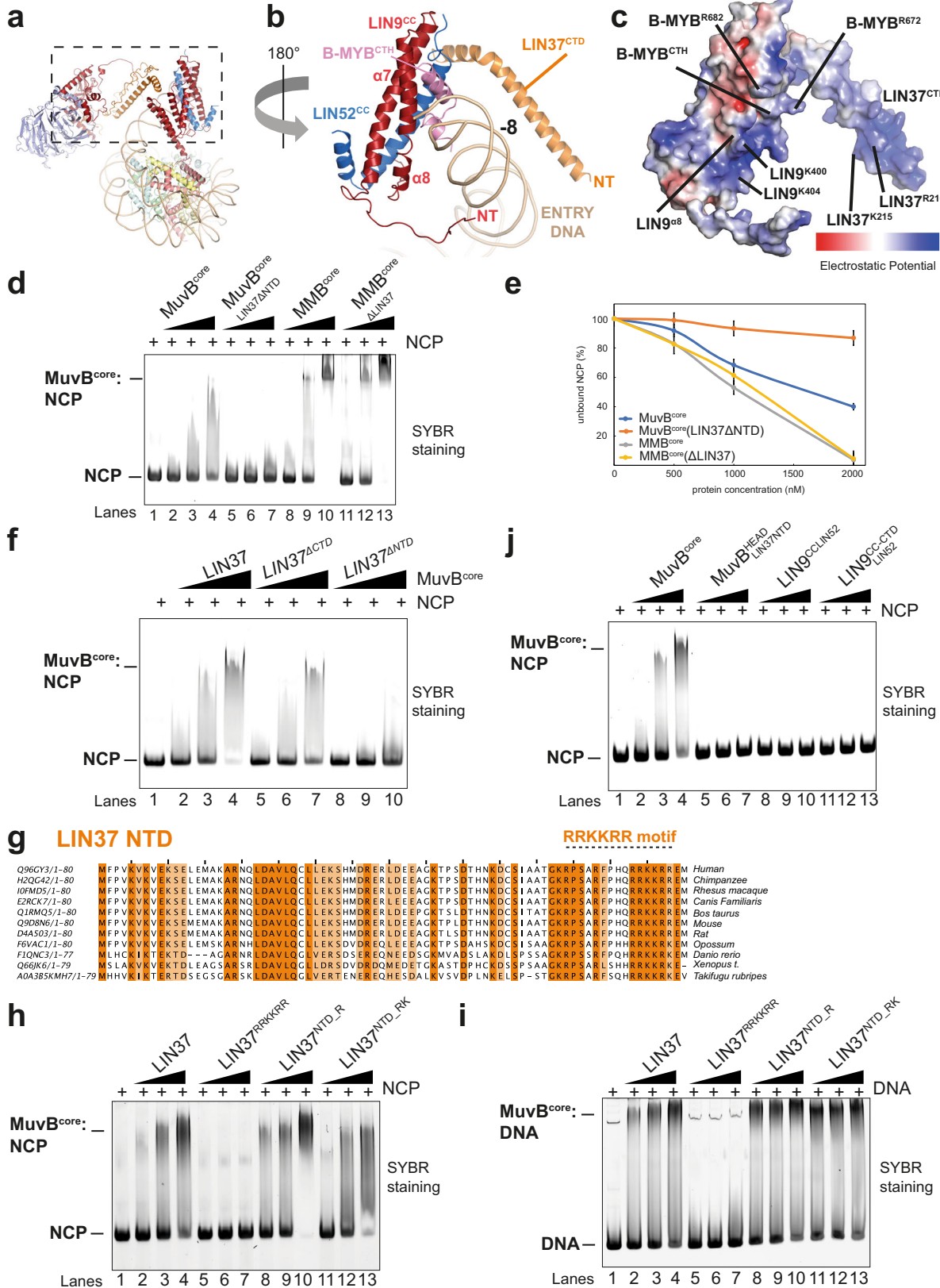

several cross-links are observed between LIN37[CTD] and LIN9 α5 and 6, thereby supporting the fact that these portions of the complex are in proximity to each other.

LIN37[CTD] contains many basic residues which face the nucleosomal entry DNA of the MMB[core]:nucleosome structure (Figs. 1b, c, 2e, f, 3a–c), suggesting that this domain is involved in anchoring the MuvB[HEAD] onto

the outer side of the nucleosomal entry DNA. Consistent with this, a MuvB[core] complex reconstituted without LIN37[CTD] (ΔLIN37CTD mutant) binds the nucleosome with lower affinity than the control MuvB[core] (Fig. 3f, lanes 5–7 versus 1–4).

In conclusion, RbBP4 subcomplex is formed at RbBP4 blade 1, and 6 where LIN9 DIRP and tudor bind, and this assembly is stabilised by

**Fig. 3 | MuvB^TAIL complex forms a V-shaped channel with MuvB^HEAD for the nucleosomal entry DNA. a, b** close-up view from **a** on the V-shaped protein channel formed by B-MYB^CTH:LIN9CC:LIN52CC and LIN37^CTD. For clarity, LIN37^CTD loops and the LIN9^CTD are not shown. **c** This channel, which contains basic residues (indicated) clamping the entry DNA, is here represented as electrostatic surface potential representation (−/+5.000). **d, e** Structure based-mutational analysis to functionally validate interactions between LIN37, B-MYB, and the nucleosome in EMSA. Either MuvB^core or MMB^core containing LIN37 or not (ΔLIN37), or containing a LIN37 NTD truncation (LIN37ΔNTD), were assayed for their ability to bind NCP 167 (at a molar ratio of 1:1, 1:2, 1:4 NCP:MuvB). Quantifications for **d** are plotted in (**e**). Data are presented as mean values ± standard error of the mean of three independent experiments (n = 3). **f** MuvB^core containing FL-LIN37, or its NTD truncation (LIN37ΔNTD), or its CTD truncation (LIN37ΔCTD), were assayed as in (**d**). This experiment was repeated independently three times with similar results. **g–i** Sequence alignment showing conservation on the LIN37 NTD. The RRKKRR motif region mutated in the assays in **h** and **i** is indicated. **h, i** MuvB^core containing LIN37, or LIN37 Arg to Glu point mutants i.e., R64E/R68E/R73E/R74E/R77E/R78E (RRKKRR), R19E/R37E (NTD_R) and R19E/R37E/K5A/K7A/K10A/K32A/K46A/K54A (NTD_RK), were assayed as in **d** for either nucleosome (**h**) or DNA (**i**) binding. These experiments were repeated independently three times with similar results. **j** EMSAs with NCP 167 and either MuvB^core or MuvB^HEAD-LIN37FL or LIN9CC:LIN52 or LIN9CC-CTD:LIN52, to assay the ability of the MuvB^HEAD and the MuvB^TAIL to bind nucleosome. Molar ratios used here were 1:1, 1:2, 1:4 NCP:MuvB complex. This experiment was repeated independently three times with similar results.

LIN37 middle domain and PRL (Fig. 1a and Supplementary Fig. 9a). Binding to the nucleosome stabilises LIN37^CTD in respect to the RbBP4 subcomplex, which extends from the LIN37^PRL to the outer surface of the nucleosomal entry DNA (Fig. 1b), thereby contributing to nucleosome binding.

## MuvB^TAIL structure and assembly

MuvB^TAIL module is anchored on the nucleosomal entry DNA extending towards the nucleosomal disc (Fig. 1b, c). Strikingly, MuvB^HEAD and MuvB^TAIL form a V-shaped tunnel which clamps onto the entry DNA (Figs. 1b, c, 3b, c and Supplementary Movie 1).

Most of the MuvB^TAIL is composed of the LIN9 C-terminus including a coiled-coil domain interacting with LIN52 coiled-coil domain (hereafter named LIN9:52CC) and with the B-MYB C-terminal helix (CTH). The structure of this MMB subcomplex was previously solved by X-ray crystallography[45] and it could be fit unambiguously in our MuvB^TAIL:nucleosome cryo-EM reconstruction (Supplementary Fig. 5a, b, f and Supplementary Movie 1).

LIN9:52CC binds the entry DNA at the superhelical location (SHL) −8, located at the latest 10 base pairs at one end of the 167 nucleosomal DNA construct used. This interaction is mainly mediated by the B-MYB^CTH, which contains two arginine residues oriented towards the DNA (i.e., Arg672 and Arg682) (Figs. 1b, c and 3b, c). Consistent with this, MMB^core binds the nucleosome better than MuvB^core (Fig. 3d, compare lanes 1–4 versus lanes 8–10 and Fig. 3e).

The basic LIN37^NTD presents several crosslinks with MuvB^TAIL mainly at the LIN9:52CC module (Supplementary Fig. 6c and Fig. 3g) suggesting that this region is in proximity of the nucleosomal DNA. Therefore, we investigated if LIN37^NTD could contribute to nucleosome binding. Strikingly, deletion of this portion of LIN37 abolishes the binding of MuvB^core to nucleosomes (Fig. 3d, lanes 1–7 and Figs. 3e and 3f, lanes 1–4 vs lanes 8–10). Importantly deletion of LIN37^NTD did not alter the structural stability and the assembly of MuvB^core (Supplementary Fig. 2h). Given the striking effect of this mutation, we narrowed down the portion of LIN37^NTD that contributes to nucleosome binding and found that a sequence containing a RRKKRR motif (residues 64-78) is absolutely required by MuvB^core to bind both the nucleosome and free DNA (Fig. 3g–i, lanes 1–7). Strikingly, the same motif has DNA binding function in INCENP[46].

In summary, the MMB^core complex clamps the nucleosomal entry DNA via a protein tunnel formed by MuvB^HEAD and MuvB^TAIL (Fig. 1b, c, and 3b, c), and DNA binding depends mainly on LIN37 and B-MYB.

Importantly, both MuvB^HEAD and MuvB^TAIL are required for full nucleosome binding activity. In fact, the MuvB^HEAD-LIN37NTD construct, as well as constructs containing either a LIN9CC:LIN52 or a LIN9CC-CTD:LIN52 subcomplex, are not able to bind the nucleosome individually (Fig. 3j, lanes 1–4 versus lanes 5–13).

## MMB remodels the nucleosome

In our MMB^core:nucleosome structure, the nucleosome is remodelled. Comparison between this nucleosome and an apo nucleosome

reconstruction, isolated from our cryo-EM dataset by 3D classification (Fig. 4a–c and Supplementary Fig. 4c), shows that 20 base pairs of the entry DNA (from SHL −6 to SHL −8) are rotated 50 degrees over the nucleosomal disc and held in position by the MMB^core complex (Fig. 4a–d). This entry DNA is lifted about 25 Å upwards at its end as visible in our density. Moreover, the entry DNA is bent inwards in respect to a normal nucleosome (Fig. 4e). Strikingly, density extending from the N-terminus of LIN9CC passes under the entry DNA wedging it (Supplementary Fig. 5f and Fig. 1b, c), before continuing onto the LIN9 N-terminus in the MuvB^HEAD module. This suggests that the MMB complex topologically entraps the nucleosomal exit DNA.

On the other side, -10 base pairs of the exit DNA at SHL +7 are not visible in our complex (Fig. 4f), indicating that this stretch of DNA is disordered. Interestingly, the unwrapping at SHL +7 is also observed in structures of ATP-dependent chromatin remodelling complexes with the nucleosome[47,48].

Out of 167 bp, only 147 bp are visible in our MMB^core:nucleosome structure. Consistently with this, MNase digestion assay of the nucleosome in the presence of MMB^core shows the stabilisation of a 147 bp fragment (Fig. 4g).

Structural comparison of the MMB^core:nucleosome complex, with the histone H1:nucleosome complex[49] shows that simultaneous MMB and H1 binding on the same nucleosome would be incompatible (Fig. 5e and h). This is because, firstly, similar to the linker H1 histone, MuvB^HEAD localises near the nucleosome dyad (Fig. 4b, h); second, the path of the exit DNA is heavily rearranged in MMB^core:nucleosome complex, and not compatible with the formation of a H1:nucleosome complex (Fig. 4d, e and h). Consistent with this, adding H1 to a MMB^core:nucleosome complex reduces the amount of MMB^core:nucleosome, while the H1:nucleosome complex forms (Fig. 4i).

In conclusion, the MMB^core complex binds the entry DNA thereby distorting it, leading to the exposure of about 10 base pairs of nucleosomal DNA at and underneath the +/−7 SHL locations. We hypothesise that this exposed DNA, can be accessible to downstream TFs and can also facilitate the assembly of the transcriptional machinery, as discussed further below.

## MMB^core interactions with histones and modelling of a full MMB:nucleosome complex

It has been shown that RbBP4, when bound to LIN9 and LIN37, is able to bind the H3 histone tail[9]. In our structure, RbBP4 is in proximity of the histone H3 tail (Fig. 1c), indicating that the binding mode of MMB with the nucleosome would allow interactions between RbBP4 and the H3 tail. However, our cryo-EM data do not resolve such an interaction directly. In general, histone tails are not required for MuvB^core:nucleosome complex formation (Supplementary Fig. 3d). Also, we cannot see any effect when adding the H3K4me3 modification, which is present at MuvB target promoters[12], on MuvB^core: nucleosome complex formation (Supplementary Fig. 3e).

On the other hand, our cryo-EM maps show density for the conserved LIN9^CTD which emerges from the C-terminal end of the LIN9CC

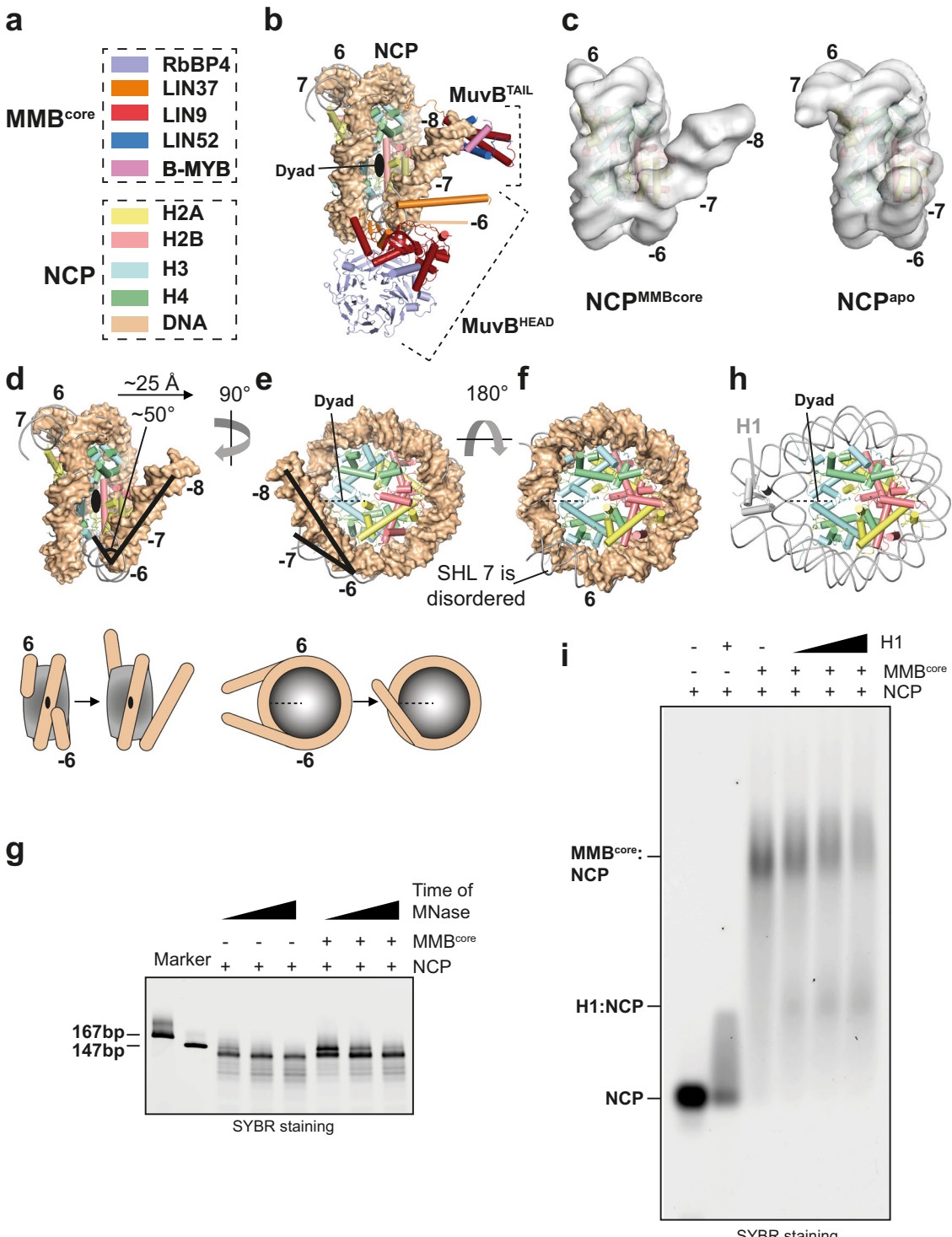

**Fig. 4 | MMB initiates chromatin remodelling. Details of distortions at the entry and exit DNA sites.** Dyad view of the MMB^core:NCP structure is shown in (**b**), colour coded as specified in (**a**). The LIN9^CTD is not shown for visualising better the distorted nucleosomal DNA. **c** cryo-EM density is displayed for the NCP complex within either the MMB^core:NCP (Supplementary Fig. 5g) or the apo NCP (Supplementary Fig. 4c). This comparison shows the conformational change at −/+7 SHL locations. **d**–**f** The DNA distortions are visualised in detail with different views of the NCP model (DNA in surface representation) from the MMB^core:NCP structure which is superposed to an apo NCP model (DNA in ribbon representation). **g** MNase digestion profile of NCP167 in the absence and presence of MMB^core. This experiment was repeated independently three times with similar results. **h** Cartoon illustration of nucleosome:H1 complex (PDB-ID: 5NI0) **i** EMSA experiment shows H1 titration reduces NCP association with the MMB^core. This experiment was repeated independently three times with similar results.

domain (Supplementary Figs. 5f, h, 10d, Fig. 1b, c, and Supplementary Movie 1). This domain is composed of a 4-helix bundle where the longest C-terminal helix (α12) extends onto the nucleosome disc until its very C-terminus docks into the nucleosome acidic patch and the

neighbour H4 histone (Fig. 5a, b). Consistently, increasing concentrations of the acidic patch binder LANA peptide[50], can compete with MuvB for nucleosome binding, even in presence of the CHR element (Fig. 5c).

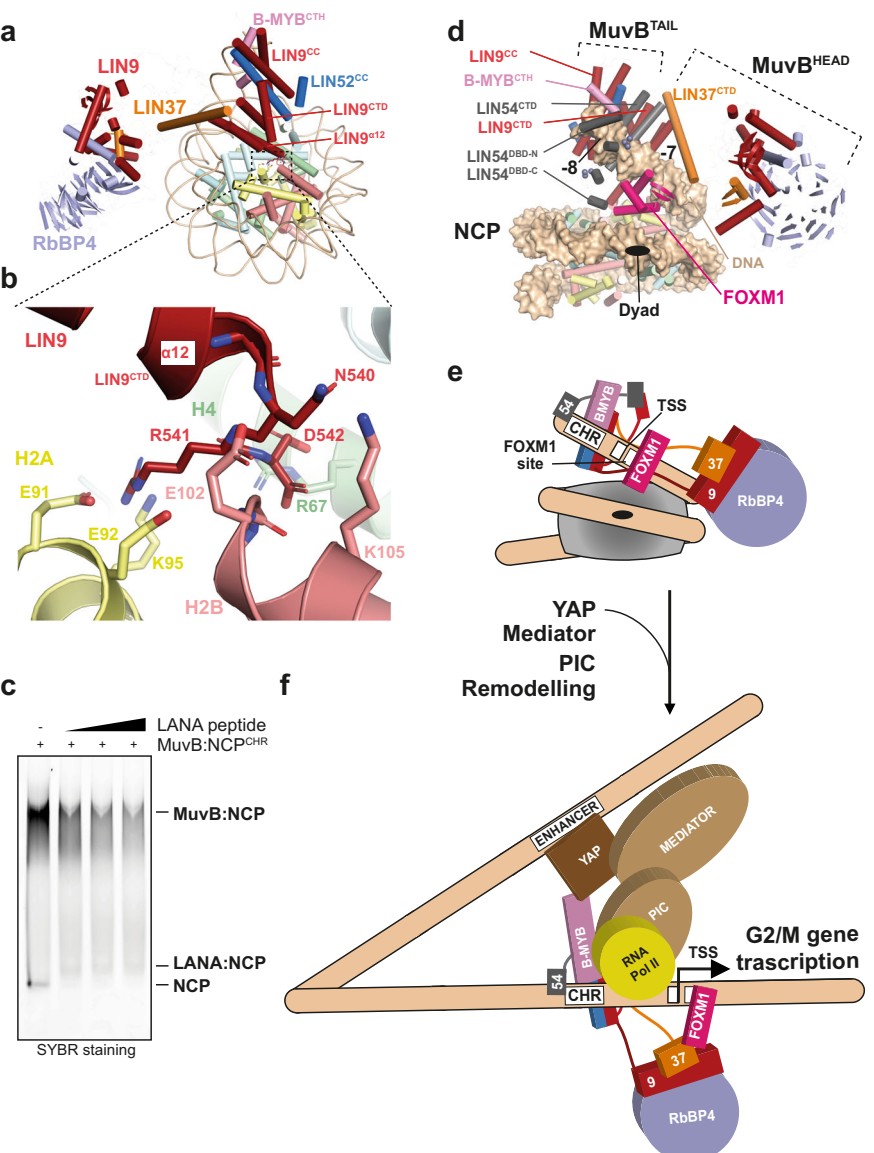

**Fig. 5 | Interactions of MMB^core with histones and model of pioneer transcription factor function of the MMB complex.** The last helix of LIN9 (α12) interact with the nucleosome acidic patch and histone H4. A zoomed view of **a** is shown in (**b**). Although the resolution of our structural data in this area does not allow for resolving side-chains, possible side-chains interactions between the LIN9 C-terminal NRD sequence and residues from H2A and H2B histones (acidic residues are part of the nucleosome acidic patch) and H4 are modelled based on docking of secondary structure elements in the NCP:LIN9^CTD map (Supplementary Fig. 5f, h) and here depicted. **c** LANA peptide competes with MuvB for binding the nucleosome. A complex of MuvB with a nucleosome containing a CHR sequence was run in an EMSA in either the absence or the presence of increasing concentrations of LANA peptide. This experiment was repeated independently three times with similar results. **d** Modelling of a MMB-FOXM1:nucleosome complex. The AlphaFold model from Supplementary Fig. 6d was superposed to the LIN9^CTD. The interaction with the CHR element was modelled by superposing the DNA from PDB-ID: 5FD3, into the SHL −8 region of our MMB^core:nucleosome complex. FOXM1 was modelled by superposing the DNA from PDB-ID: 7FJ2 into the SHL −7 region of our MMB^core:nucleosome complex. The positioning of FOXM1 at the entry DNA is suggested by nucleosomal binding preferences of FOX TFs related to FOXM1[67]. **e** Schematic cartoon of the complex in (**d**). Protein loops in **a** and **d** are hidden for reducing the complexity of the illustration. **e, f** A model of pioneer transcription factor function of the MMB complex is illustrated. CHR sites are located right upstream of a nucleosome, and they are exposed as parts of nucleosome depleted regions (NDRs)[7,68]. During S phase when MMB forms, remodelling of a nucleosome by the pioneer transcription factor function of MMB would allow exposure of *cis* acting sequences, including the TSS. This would allow establishment of competency for transcriptional activation at cell cycle genes. This, combined with the recruitment of downstream factors, and ATP-dependent chromatin remodellers, could promote formation of the preinitiation complex (PIC) and recruitment of the RNA polymerase. MMB can also participate in promoter–enhancer interactions[56,57]. Full activation of G2/M genes is a very complex process, which requires post-translational modifications on both B-MYB and FOXM1[20,27].

Moreover, our crosslinking data suggests that the LIN9^CTD interacts with the LIN54^CTD (Fig. 1a and Supplementary Fig. 6d). We used AlphaFold[51] for modelling this interaction (Supplementary Fig. 6d–f) and thereby obtaining a complete model of the full MMB:nucleosome complex (Fig. 5d). This modelling exercise suggests that LIN54^CTD may occupy a cavity between LIN9^CTD, the LIN9:52CC, and the DNA. This arrangement would place the DNA binding domains of LIN54 in proximity of a CHR element at the SHL −8 of the nucleosome (Figs. 1a and 5d). LIN54sh crosslinks with all the other MuvB subunits (Supplementary Fig. 6e), thereby supporting our modelling that LIN54 is present at the centre of MuvB between TAIL and HEAD modules.

In conclusion, our data suggest that LIN9^CTD is a critical component of the MMB complex, which connects MuvB directly to the nucleosome acidic patch and to the CHR via recruiting LIN54.

## Discussion

Our cryo-EM structure shows that the MMB complex consists of two main modules, here named MuvB$^{HEAD}$ and MuvB$^{TAIL}$. These two modules are flexibly linked by the main scaffolding protein of the complex LIN9 (Fig. 1b, c and Supplementary Movie 1). LIN9 orchestrates the assembly of MuvB$^{HEAD}$ via its N-terminal DIRP and tudor domains and the assembly of MuvB$^{TAIL}$ via its C-terminal coiled-coiled domain, which recruits LIN52CC and B-MYB$^{CTH}$. LIN9 together with RbBP4 recruits LIN37 via its middle domain at the MuvB$^{HEAD}$. LIN37 is a stabilising factor since it buttresses LIN9 on RbBP4 and also, we show that LIN37 within the MuvB complex is a DNA-binding protein. The LIN37 termini are essential for full nucleosome binding ability of the MuvB complex. Strikingly, these regions become less important for nucleosome binding in presence of B-MYB. In fact, the MMB$^{core}$ complex can still bind a nucleosome in the absence of LIN37 (Fig. 3d). Our structure shows an explanation for this. B-MYB$^{CTH}$ recruited by LIN9:52CC directly contacts the entry DNA upstream of LIN37 (Figs. 1b, 3b, c and Supplementary Movie 1). This is consistent with prior data showing that B-MYB C-terminus is sufficient for B-MYB-dependent transcriptional activation[52,53] and that LIN37 is dispensable for the transcriptional activating functions of MuvB[44,54].

Our MMB:nucleosome complex also shows that MMB binds the nucleosome with a 1:1 stoichiometry and initiates chromatin remodelling, which is a functional feature of pioneer TFs. By clamping the entry DNA, MMB lifts 20 base pairs thereby exposing DNA at SHL −7. The DNA following the distortion is not visible in other pioneer TFs complexes structures[34,35]. In our structure, this region is visible because it is clamped by the MMB complex itself.

How is this remodelling achieved? Such a conformational change requires disruption of interactions between histone H3 with SHL −7 (Fig. 4d). Similarly to other pioneer TFs, in the case of MMB, disruption of histone:DNA interactions may be compensated by the energy of binding between MMB and the nucleosomal DNA. In addition, RbBP4 is heavily negatively charged, and it is positioned in proximity of SHL −6 (Fig. 4b), where the DNA distortion starts. Based on this, an intriguing hypothesis emerges whereby a charge repulsion mechanism between RbBP4 and the DNA at SHL −7 could contribute to DNA path deviation. This could also explain the high mobility of MuvB$^{HEAD}$ in respect to the nucleosome as observed in our cryo-EM data (Supplementary Fig. 4f). Moreover, we observe density coming from LIN9CC N-terminus directed towards MuvB$^{HEAD}$ (Fig. 1c and Supplementary Fig. 5f) This suggests that MMB topologically entraps the entry DNA, thereby contributing to hold the DNA in position.

The conformation of the entry/exit DNA in our MMB:nucleosome complex structure would not be compatible with linker histone H1 binding. Consistently, we show that H1 can compete MMB from the nucleosome (Fig. 4e and 4h, i). This is consistent with the fact that CHR-containing regions have typical features of actively transcribing genes[12] and not of heterochromatin regions, which are enriched in H1[55].

Importantly, LIN54 and the MuvB target CHR sequence are not included in our MMB:nucleosome structure, indicating that the nucleosome-dependent remodelling by the MMB complex is independent of CHR recognition and so the remodelling would be an intrinsic property of the core MMB complex. Furthermore, we observe that the last helix of LIN9 is inserted into the acidic patch of the nucleosome (Fig. 5a, b). This suggests that a direct interaction of the MMB complex with the nucleosome contributes with anchoring and positioning the complex onto the nucleosome. The ability of binding both nucleosomal DNA and the histone octamer could explain why MuvB stabilises nucleosomes[9].

Considering previous and our current work, we propose a model of MMB function as a pioneer TF complex (Fig. 5d–f). MuvB disassembled from DREAM at the G1/S phase transition assembles with B-MYB, thereby forming the MMB complex (Fig. 5d, e). The MMB complex is located at CHR DNA elements, in proximity of the TSS and the +1 nucleosome. This nucleosome is remodelled by MMB thereby exposing DNA at −/+7 SHL locations (Fig. 5d, e).

This would facilitate the recruitment of downstream transcription factors (i.e., FOXM1)[20], and the assembly of the basal transcriptional machinery at the TSS. MMB has been shown to participate in interactions between the basal promoter and enhancer elements[56,57], which underpins the importance of this complex in transcriptional activation of mitotic genes.

In conclusion, our structure-function investigation of the MuvB complex gives insights on complex assembly, interactions, and binding mode with a nucleosome. Our study paves the way for the investigation of a mechanistic model of MuvB function in transcriptional activation and B-MYB mediated oncogenesis.

## Methods

### Protein expression and purification

The codon optimised coding sequences (CDSs) of the wild-type, truncations, and point-mutants of human MuvB and B-MYB proteins were obtained from Genscript or GeneArt (Thermo Fischer Scientific) and cloned into pFASTBAC-1 or pACEBAC-1 vector using standard methods. Construct boundaries are as follows: LIN9$^{FL}$(1–542), LIN9$^{MuvB(HEAD)}$ (94–278), LIN9$^{CC}$(325–450), LIN9$^{CC-CTD}$(332–542), LIN37$^{FL}$(1–246), LIN37$^{ΔNTD}$(92–246), LIN37$^{ΔCTD}$(1–132), RbBP4$^{FL}$(1–425), LIN52$^{FL}$(1–116), LIN54$^{CTD}$(635–749), LIN54$^{short}$(515–749), B-MYB$^{FL}$(1–700).

Point-mutants in LIN37$^{FL}$: LIN37$^{RRKKRR}$(R64E/R68E/R73E/R74E/R77E/R78E), LIN37$^{NTD-R}$(R19E/R37E), LIN37$^{NTD\_RK}$(R19E/R37E/K5A/K7A/K10A/K32A/K46A/K54A). For MuvB complex expression and assembly, Hi-5 cells at a density of $1.5 \times 10^6$ cells ml$^{-1}$ were co-infected with pre-cultures of Sf9 cells each pre-infected with recombinant MuvB baculoviruses. Cells were harvested at a viability of ~85% which typically took approx. 48 h and stored at −80 °C until processing. B-MYB protein was expressed separately in a similar manner to MuvB.

Hi-5 cells expressing MuvB (sub-)complexes (wild-type or mutants) were resuspended in buffer containing 50 mM Tris pH8, 300 mM NaCl, 0.5 mM TCEP, 5% Glycerol, 1 mM EDTA, 0.1 mM PMSF, 2 mM benzamidine, 5 U ml$^{-1}$ benzonase (Novagen), Complete EDTA-free protease inhibitors (Roche) and lysed by sonication followed by centrifugation. The filtered soluble fraction of the lysate was loaded on a Strep-Tactin Superflow (30060, Qiagen) column for purification using the StrepII$^{x2}$ tag on LIN37 or RbBP4 (the latter on the MuvB ΔLIN37 complex). Bound protein was washed with 20 CV with buffer containing 50 mM Tris pH8, 300 mM NaCl, 0.5 mM TCEP, 5% Glycerol, followed by a wash with 4 CV 50 mM Tris pH8, 1000 mM NaCl, 0.5 mM TCEP, 5% Glycerol and subsequently eluted with buffer containing 50 mM Tris pH8, 300 mM NaCl, 0.5 mM TCEP, 5% Glycerol, 2.5 mM d-Desthiobiotin. Strep tag was removed by a 4 °C o/n incubation with 3 C protease. Final polishing was achieved by SEC on a S200 or Superose 6 column (GE Healthcare) in buffer containing 20 mM HEPES pH 8, 150 mM NaCl, 0.5 mM TCEP. Peak fractions were pooled and concentrated up to 5 mg/ml.

B-MYB protein was purified separately as per above protocol. For the MMB complex assembly, freshly prepared MuvB:B-MYB proteins were mixed at a 1:4 ratio and incubated on ice for 1 h followed by SEC on a Superose 6 column in buffer containing 20 mM HEPES pH 8, 100 mM NaCl, 0.2 mM TCEP.

Recombinant H1 with N-terminal Hexahistidine tag and C-terminal StrepII$^{x2}$ tag was expressed in *Escherichia coli*. Protein expression was induced by adding 0.5 mM IPTG when OD600 reached 0.7, and allowed to continue overnight at 18 °C. Cells were lysed by sonication. H1 protein was purified using Strep-Tactin Superflow Plus (QIAGEN) and HiTrap Heparin HP (GE Healthcare) according to standard protocol. Both N-terminal and C-terminal tags were removed by overnight 3C protease digestion. Untagged H1 protein was passed to a S200 16/600 column (GE Healthcare). Pure H1 fractions were concentrated, flash frozen in liquid Nitrogen, and stored at −80 °C for later use.

## NCP167 assembly

Recombinant human histones H2A, H3, H4, and *Xenopus laevis* H2B were overexpressed separately in *Escherichia coli*, purified from inclusion bodies, and assembled together into a histone octamer as described previously[58]. Briefly, the inclusion bodies for each histone were resolubilised in Urea-containing buffer individually and further purified by ion exchange column. Denatured histones were mixed at a H2A:H2B:H3:H4 molar ratio of 1.2:1.2:1:1, and dialysed against refolding buffer containing 2 M NaCl, 10 mM Tris-HCl pH 7.5, 1 mM EDTA, and 10 mM β-mercaptoethanol. Refolded histone octamer was purified from aggregates and excess H2A-H2B dimer using gel filtration on a S200 16/600 column (GE Healthcare), run in the same refolding buffer. Pure histone octamer fractions were concentrated, mixed to a final glycerol concentration of 50% (v/v), and stored at −20 °C for later use.

Nucleosomal DNA was prepared using Plasmid Giga Kit (QIAGEN). Plasmid containing four repeats of 167 bp sequence (TAC CCG GGA TAT CGA GAA TCC CGG TGC CGA GGC CGC TCA ATT GGT CGT AGA CAG CTC TAG CAC CGC TTA AAC GCA CGT ACG CGC TGT CCC CCG CGT TTT AAC CGC CAA GGG GAT TAC TCC CTA GTC TCC AGG CAC GTG TCA GAT ATA TAC ATC CGA TAT CCC GGG TA) was transformed into XL10 Gold bacteria. Cells were grown overnight, and plasmid extraction was done according to QIAGEN Plasmid Giga Kit protocol. Extracted plasmid was digested with BstZ17I (NEB) overnight. The 167 bp nucleosomal DNA was purified from vector DNA by ion exchange column. Trace amount of BstZ17I was inactivated by phenol-chloroform extraction.

Nucleosomal DNA containing CHR sequence was prepared by Polymerase Chain Reaction (PCR). The PCR was performed according to standard protocol using Taq DNA Polymerase with ThermoPol® buffer (NEB). Widom 601 sequence was used as a template, and the PCR reaction was scaled up to 20 ml reaction volume. PCR product was purified using HiTrap Q HP anion exchange column (GE Healthcare). Fractions containing pure nucleosomal DNA were pooled, ethanol precipitated, dissolved in water, and finally stored at −20 °C for later use. The sequence of Nucleosomal DNA containing CHR sequence is GAG TTC AAA ATC GAG AAT CCC GGT GCC GAG GCC GCT CAA TTG GTC GTA GAC AGC TCT AGC ACC GCT TAA ACG CAC GTA CGC GCT GTC CCC GCG TTT TAC CGC CAA GGG ATT ACT CCC TAG TCT CCA GGC ACG TGT CAG ATA TAT ACA TCC GAT.

Nucleosome (NCP167 and NCP containing the CHR sequence) reconstitution was performed as described previously[58] with minor modifications. Briefly, histone octamer and nucleosomal DNA were mixed at 1.1:1 ratio in a high-salt reconstitution buffer containing 2 M NaCl, 20 mM HEPES pH 7.5, 5 mM DTT, 1 mM EDTA. Sample was transferred into a dialysis button (Hampton Research), and the button was placed inside a beaker containing the same high-salt reconstitution buffer. Low-salt reconstitution buffer containing 20 mM NaCl, 20 mM HEPES pH 7.5, 5 mM DTT, 1 mM EDTA was slowly pumped into the dialysis beaker to gradually reduce the salt concentration and allow NCP formation. Sample was transferred to a fresh low-salt reconstitution buffer after 24 h and allowed to further equilibrate for 4 h. Soluble NCP167 was separated from aggregates by centrifugation, and stored at 4 °C for subsequent use.

The NCP167 with the H3K4me3 modification was purchased from ActiveMotif.

## Tailless NCP167 preparation

Reconstituted NCP167 was diluted in Digestion Buffer containing 20 mM HEPES pH 7.5, 75 mM NaCl, 1 mM TCEP to a final concentration of 1.8 µM. TPCK-trypsin immobilised on magnetic beads (Takara) was washed and equilibrated in the same Digestion Buffer, and mixed with NCP167 at 1:1 volume ratio. Digestion was stopped after 20, 40, 60, and 80 min by removing the magnetic bead-coupled trypsin from the solution. All samples were tested on SDS-PAGE to assess the completion of histone tails removal.

## MNase protection assay

The NCP was mixed with MMB^core at a concentration of 0.6 and 1.8 µM, respectively and topped up with EMSA buffer containing 20 mM HEPES pH 7.5, 50 mM NaCl, 0.5 mM TCEP to final volume of 10 µL. As a control, a reaction containing NCP alone was also prepared separately. MNase reaction buffer (NEB) was added to each sample at a final concentration of 1×. 2.5 µL of 1000 unit/µl MNase (NEB) was then added to each sample. Reaction was conducted at room temperature. 3 µL were aliquoted from each sample at 30 s, 1 min, and 1.5 min after the MNase addition. Reaction was terminated by adding Proteinase K solution (ThermoFisher Scientific) into each aliquot. Samples were analysed on Novex™ 4–12% TBE gel (ThermoFisher). The gels were stained with SYBR Safe and scanned using Typhoon FLA 9500 imager.

## SEC-MALS

Analytical SEC-coupled with multi-angle laser light scattering (MALS) was performed on an Agilent 1260 Infinity II HPLC system. Light scattering and differential refractive index (dRI) profiles of the samples were analysed in-line using the DAWN and Optilab instruments from Wyatt Technologies. Samples of purified MuvB complex at different concentrations (20 µL volume) were applied to a Superose 6 3.2/300 column (GE Healthcare) equilibrated with 20 mM HEPES pH 7.5, 150 mM NaCl, 0.5 mM TCEP at a flow rate of 0.05 ml/min. The data were analysed according to the Zimm light scattering model and a dn/dc of 0.185 mL/g on software ASTRA 7.3.1 (Wyatt).

## Mass photometry

Mass photometry experiments were performed on a Refeyn One^MP mass photometer (Refeyn Ltd, Oxford, UK) as described previously[59]. Microscope coverslips (630-2187, VWR) and silicone CultureWell™ gaskets (GBL103250, Sigma-Aldrich) were cleaned with 100% iso-propanol and water and dried with compressed air. The gaskets were placed on top of the cleaned coverslips on the sample stage of the mass photometer as per manufacturer's instructions.

All measurements were performed at least five times independently, in separate wells, in buffer containing 20 mM HEPES pH 8, 150 mM NaCl, 0.5 mM TCEP. Firstly, 12 µL buffer was pipetted into a gasket-well followed by focal point acquisition using the autofocus functionality in the Refeyn Acquire^MP 2.3.1 software. Secondly, 2 µL of 500 nM of MuvB complexes (diluted to this concentration just prior to measurement to avoid complex disassembly) was pipetted in the buffer-containing gasket-well and mixed to observe individual landing events below saturation levels. Mass photometry movies of 6000 frames were recorded from a 10.8 × 10.8 µm instrument field of view. Data were processed using the default pipeline on the Refeyn Discover^MP 2.3.0 software. Individual particle contrasts from each Video were converted to mass using a contrast-to-mass (C2M) calibration which was performed in the acquisition buffer. Data were plotted as normalised histograms with a bin width of 4.4 and fitted to a Gaussian peak. For the calibration, 2 µL of a 1:100 pre-diluted NativeMark unstained protein standard (LC0725, Thermo Scientific) was added to an acquisition gasket-well; the 66, 146, 480, 1048 kDa masses were used for a standard calibration curve in the Discover^MP software.

## Cross-linking of MuvB-apo for cryo-EM

30 µL of freshly purified MuvB complex at a concentration of 28 µM was incubated with 0.1 µL 25% Glutaraldehyde (G5882, Sigma-Aldrich) for 2 min on ice followed by quenching with 2 µL 500 mM Tris pH8 and immediate SEC on a Superose 6 3.2/300 column (GE Healthcare) in buffer containing 20 mM Tris pH 8, 100 mM NaCl, 0.2 mM TCEP. Eluted fractions were analysed by SDS-PAGE and MuvB monomer-containing fractions were pooled and concentrated to 20 µL and re-injected on the SEC column as described above to remove any higher-order contaminants. Monomeric-containing fraction was used immediately for cryo-EM grid preparation.

## Gradient fixation (GraFix)

NCP167 and MMB[core] were mixed at 1:2.5 molar ratio and incubated at 4 °C for at least 30 min. GraFix was done as described previously[41]. A continuous gradient containing 20 mM HEPES pH 7.5, 50 mM NaCl, 0.1 mM TCEP, 10–30% sucrose, 0–0.1% glutaraldehyde was generated using a Gradient Master (Biocomp). Samples were centrifuged using a SW60 Ti ultracentrifuge rotor at 165, 100 × $g$ for 16 h at 4 °C. Samples were manually fractionated by hand, and the crosslinking was terminated by adding Tris-HCl pH 7.5 to a final concentration of 35 mM. GraFix fractions were analysed on a 5% native PAGE, and those containing crosslinked NCP167-MMB[core] complex were concentrated, and buffer exchanged into 10 mM Tris-HCl pH 7.5, 50 mM NaCl, 0.1 mM TCEP.

## Cryo-EM grid preparation

**MuvB-apo complex.** 2 μL of the cross-linked monomeric MuvB complex was applied to Quantifoil R1.2/1.3 Cu 300 grids which were glow discharged using Easiglow (Pelco) at 15 mA for 1 min. After sample application, the grids were immediately blotted for 5 s at 4 °C and 100% humidity and plunged into liquid ethane using a Vitrobot mark IV (Thermo Fisher).

**MMB[core]:NCP complex.** 2 μL of the GraFix-treated sample was applied to Quantifoil R1.2/1.3 Cu 300 grids which were glow discharged using Easiglow (Pelco) at 15 mA for 1 min. The grids were incubated for 5 s and then blotted (blot force 3) for 5 s at 4 °C and 100% humidity and plunged into liquid ethane using a Vitrobot mark IV (Thermo Fisher).

## Cryo-EM data collection and processing

**MuvB-apo complex.** We collected 10,183 movie stacks in EER format on a Glacios (Thermo Fisher) Cryo-TEM 200 kV equipped with a Falcon 4 detector, based at ICR, at a nominal magnification of 240,000, which yielded a pixel size of 0.567 Å per pixel. These movies were live-processed during collection using the cryoSPARC live programme[60].

Movie stacks were frame aligned and binned 4 times giving a pixel size of 2.268 Å per pixel. 8820 Images with resolution better than 6 Å and a total motion of 30 pixels (estimated during frame alignment) were selected for further processing. A blob picker was used for initial particle picking during the live processing, and for obtaining initial 2D class averages. Templates generated from the latter, were used for template picking and TOPAZ[61] training and picking with cryoSPARC. In total, 1,184,371 particles were picked; duplicated particles were removed by using the "remove duplicates" function implemented in cryoSPARC. This step removed doubly picked particles within 60 Å (shortest dimension of the MuvB particles is -50 Å). These particles were cleaned further with another 2D classification step (Supplementary Fig. 7). These particles were piped in the "ab initio model" function implemented in cryoSPARC. Three classes were used for allowing sorting of particles in the process. This generated one class containing 250,174 particles with clear domain structures and with resolved helices. 2D classification of this class showed improvement on the signal-to-noise of the class averages. These particles were then imported in RELION-3.1.1[62] by using pyem and the csparc2star.py script by Daniel Asarnow: Daniel Asarnow, Eugene Palovcak, & Yifan Cheng (2019). asarnow/pyem: UCSF pyem v0.5 (v0.5). Zenodo. https://doi.org/10.5281/zenodo.3576630. Refinement and further map improvement was performed as described in[63] and in Supplementary Fig. 7. This yielded a map at the resolution of 3.5 Å.

**MMB[core]:NCP complex.** We collected 22,293 Movie stacks in MRC format on a Titan KRIOS 300 kV cryo-TEM at eBIC (Diamond light source, BI21809-34 and BI21809-35 sessions), equipped with a K3 detector operated in super-resolution bin 2x mode, at a nominal magnification of 165 kx, which yielded a pixel size of 0.513 Å per pixel. Processing of these data was performed similarly to the MuvB-apo

complex with variations explained below and in Supplementary Fig. 4 and 5.

2D class-averages show a clear additional density protruding from the nucleosome (Supplementary Fig. 4E). Ab initio reconstruction followed by 3D classification allowed removal of partially disassembled nucleosomes and apo nucleosomes particles from a MMB[core]:nucleosome complex reconstruction (Supplementary Fig. 4C). Further 3D classification of this class shows high mobility of a portion of MMB[core] in respect to the nucleosome (Supplementary Fig. F). We named this highly mobile portion of the complex MuvB[HEAD]. Out of four 3D classes, one presented clearer density in both MuvB[HEAD] and another portion of the MMB[core] complex, here called MuvB[TAIL], which is embedded in the nucleosome disc (Fig. 1B and Supplementary Fig. 4F). 3D refinements focusing on MuvB[HEAD] and MuvB[TAIL], allowed us to obtain reconstructions at resolutions of 7.5 and 8.7 Å respectively (Supplementary Fig. 5A–D, J). For structural studies on the MuvB:nucleosome interactions we collected additional 29,102 Movie stacks. Particle data coming from the MMB[core]:nucleosome class obtained after 2D and 3D classification within this second dataset were merged with the first dataset at the stage indicated in Supplementary Fig. 4C. Additional 3D classification yielded 237,061 particles of MMB[core]:nucleosome complex (Supplementary Fig. 5 and J). Further focused 3D refinements were used to obtain MMB[core]:DNA, NCP147, NCP:LIN9CTD, and NCP127 maps (Supplementary Fig. 5E–J).

## Model building

The PDB 4PC0 was used as initial template for building RbBP4 in the MuvB-apo structure in Coot[64]. LIN9 DIRP, Tudor domain and LIN37 middle domain and proline-rich loop were build de novo based on the excellent quality of our cryo-EM map. Structural model refinement was performed with PHENIX real-space refinement[65] at the resolution of 3.5 Å.

This structure was rigid-body fit in the MuvB[HEAD] complex map (Supplementary Fig. 5B) where an additional kinked helix belonging to the LIN37[CTD] (residues 203–246, predicted by AlphaFold[51]) could be fitted, this was also guided by our XL-MS data (Fig. 3b). PDB 6C48, was fitted into the MuvB[TAIL-NCP] map (Fig. c and d). The connection of MuvB HEAD and TAIL via the LIN9 loop is modelled for showing the connectivity between these modules. PDB 6PWE was used as initial template to model the NCP-DNA complex.

AlphaFold model of the LIN9[CTD]:LIN54[CTD] subcomplex was obtained by submitting LIN9 residues 421–542 and LIN54 residues 635–749 to AlphaFold[51]. The prediction of this subcomplex has an overall a high score as judged by the predicted local-distance difference test.

## Electrophoretic mobility shift assay

NCP or DNA was mixed with MuvB or MMB complex at a 1:0 (input control) or 1:1, 1:2, 1:4 ratio unless otherwise stated. The concentration of NCP or DNA was constant at 0.5 μM. Reactions were topped up with buffer containing 20 mM HEPES pH 8, 50 mM NaCl, 0.5 mM TCEP and incubated on ice for 30 min. After the addition of 5% (v/v) sucrose, samples were resolved by gel electrophoresis using either 5% TBE-Polyacrylamide gel or 1% TBE-agarose gel. The gels were stained with SYBR Safe and scanned using either a Typhoon FLA 9500 imager or a Biorad ChemiDoc system. Quantification was performed using ImageLab (Biorad). Band intensities of unbound either NCP or DNA were normalised relative to the intensity of the NCP or DNA band in absence of the protein complex studied and plotted as a percentage of unbound NCP/DNA fraction. Means ± standard error of the mean of at least three independent experiments are shown.

## H1 competition assay

NCP167 was mixed with MMB[core] at 0.5 and 1.25 μM concentration, respectively. The mixture was incubated on ice for 15 min. H1 was

titrated to the mixture at final concentration of 0.25, 0.5, and 0.75 μM. After another 15 min incubation, sucrose was added to final concentration of 5% (v/v), and samples were analysed on 1% TBE-agarose gel. The gel was run at 100 V for 3.5 h at 4 °C. The gel was stained with SYBR Safe and scanned using Typhoon FLA 9500 imager.

### LANA competition assay

NCP containing CHR sequence was mixed with MuvB at 0.5 and 1.5 μM concentration, respectively. LANA peptide was titrated to the mixture at final concentration of 100, 200, and 300 μM. After 15 min incubation, sucrose was added to final concentration of 5% (v/v). Samples were analysed on 5% native PAGE gel. The gel was run at 100 V for 1.5 h at 4 °C. The gel was stained with SYBR Safe and scanned using Typhoon FLA 9500 imager.

### Crosslinking mass spectrometry analysis

Similarly to the cross-linking protocol described above for the apo-MuvB, 30 μL of freshly purified MuvB complex at a concentration of 28 μM was incubated with 0.75 μL DSSO (disuccinimidyl sulfoxide) (A33545, Thermo Scientific) pre-dissolved in DMSO at a concentration of 100 mM. Sample was incubated on ice for 10 min followed by quenching with 2 μL 500 mM Tris pH8 and immediate SEC on a Superose 6 3.2/300 column (GE Healthcare) in buffer containing 20 mM Tris pH 8, 100 mM NaCl, 0.2 mM TCEP. Eluted fractions were analysed by SDS-PAGE and appropriate MuvB monomer-containing fractions were pooled and concentrated to 20 μL and re-injected on the SEC column as described above to remove any higher-order contaminants. Monomeric-containing fraction was analysed by mass spectrometry as follows.

After the crosslinking reaction, triethylammonium bicarbonate buffer (TEAB) was added to the sample at a final concentration of 100 mM. Proteins were reduced and alkylated with 5 mM tris-2-carboxyethyl phosphine (TCEP) and 10 mM iodoacetamide (IAA) simultaneously for 60 min in dark and were digested overnight with trypsin at final concentration 50 ng/μL (Pierce). Sample was dried and peptides were fractionated with high-pH Reversed-Phase (RP) chromatography using the XBridge C18 column (2.1 × 150 mm, 3.5 μm, Waters) on a Dionex UltiMate 3000 HPLC system. Mobile phase A was 0.1% v/v ammonium hydroxide and mobile phase B was acetonitrile, 0.1% v/v ammonium hydroxide. The peptides were fractionated at 0.2 mL/min with the following gradient: 5 min at 5% B, up to 12% B in 3 min, for 32 min gradient to 35% B, gradient to 80% B in 5 min, isocratic for 5 min and re-equilibration to 5% B. Fractions were collected every 42 s, SpeedVac dried and orthogonally pooled into 12 samples for MS analysis.

LC-MS analysis was performed on the Dionex UltiMate 3000 UHPLC system coupled with the Orbitrap Lumos Mass Spectrometer (Thermo Scientific). Each peptide fraction was reconstituted in 30 μL 0.1% formic acid and 15 μL were loaded to the Acclaim PepMap 100, 100 μm × 2 cm C18, 5 μm trapping column at 10 μL/min flow rate of 0.1% formic acid loading buffer. Peptides were then subjected to a gradient elution on the Acclaim PepMap (75 μm × 50 cm, 2 μm, 100 Å) C18 capillary column connected to a stainless steel emitter with integrated liquid junction (cat# PSSELJ, MSWIL) on the EASY-Spray source at 45 °C. Mobile phase A was 0.1% formic acid and mobile phase B was 80% acetonitrile, 0.1% formic acid. The gradient separation method at flow rate 300 nL/min was as follows: for 95 min gradient from 5 to 38% B, for 5 min up to 95% B, for 5 min isocratic at 95% B, re-equilibration to 5% B in 5 min, for 10 min isocratic at 5% B. Precursors between 375 and 1600 m/z and charge equal or higher than +3 were selected at 120,000 resolution in the top speed mode in 5 sec and were isolated for CID fragmentation (collision energy 25%) with quadrupole isolation width 1.6 Th and Orbitrap detection with 30,000 resolution. Fragments with targeted mass difference of 31.9721 (DSSO crosslinker) were further subjected to CID fragmentation at the MS3 level with collision energy 35%, iontrap detection, max IT 50 ms, AGC 2 × 10⁴ and MS2 isolation window 2 Th. Two precursor groups were selected with both ions in the pair. Targeted MS precursors were dynamically excluded for further isolation and activation for 30 s with 10 ppm mass tolerance.

Identification of crosslinked peptides was performed in Proteome Discoverer 2.4 (Thermo) with the Xlinkx search engine in the MS2_MS3 mode. Precursor, FTMS and ITMS mass tolerances were 20 ppm, 30 ppm, and 0.5 Da respectively with maximum 2 trypsin missed cleavages allowed. Carbamidomethyl at C and Oxidation at M were selected as static and dynamic modifications respectively. Spectra were searched against a FASTA file containing the sequences of the proteins in the complex as well as a random selection of ~1000 Escherichia coli UniProt entries as an additional negative control of the crosslink matches. Crosslinked peptides were filtered at FDR < 0.01 using the Percolator node and decoy database search.

**Interpretation of the crosslinking mass spectrometry dataset for Model building.** Given the high flexibility between MuvB^HEAD and MuvB^TAIL (Supplementary Fig. 4F) the cross-links between these two modules were excluded from the structural interpretation (Supplementary Fig. 6B and Supplementary Data 1).

### Reporting summary

Further information on research design is available in the Nature Research Reporting Summary linked to this article.

## Data availability

The data that support this study are available from the corresponding author upon reasonable request. The structural coordinates and EM data have been deposited in the Protein Data Bank and in the Electron Microscopy Data Bank with the following accession numbers: the MuvB apo complex (including the LIN9 DIRP and tudor, RbBP4 and LIN37 MD-PRL) is deposited in PDB-ID 7R1D, EMD-14239. the MMBcore:NCP complex EM maps are deposited in EMD-15709. The mass spectrometry proteomics data have been deposited to the ProteomeXchange Consortium via the PRIDE[66] partner repository with the dataset identifier PXD031421. Source data are provided with this paper.

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

## Acknowledgements

We thank Basil J. Greber for his invaluable advice on data processing and collection of the MuvB apo complex structure. We thank Chris Richardson for the outstanding IT support. We thank Thangavelu Kaliyappan for sharing histone expression constructs. We thank Stephen Hearnshaw for his help with the mass photometry. We thank Michael Ranes and Sebastian Guettler for his help with the SEC-MALS. We thank Ruth Knight for helping with insect cell culture. We thank Alex Radzisheuskaya, Basil J. Greber, and Alessandro Vannini for critically reading and commenting on this manuscript. We thank David Barford for the discussions and for allowing C.A. to start this project in his laboratory. We thank Diamond for access and support of the cryo-EM facilities at the UK national electron Bio-Imaging Centre (eBIC), proposal BI21809-34, funded by the Wellcome Trust, MRC and BBSRC. We thank the eBIC local contact Yuewen Sheng for his great help in collecting the MMB^core:NCP datasets. We thank Christos Savva and the Leicester Institute of Structural and Chemical Biology (University of Leicester) for their great help in collecting preliminary data on this project. M.G.K. and C.A. are supported by the Sir Henry Dale Fellowship 215458/Z/19/Z. R.M. is supported by the Institute of Cancer Research (ICR), grant number allocated is GFR146X. The work of T.I.R. and J.S.C. was funded by the CRUK Centre grant with reference number C309/A25144.

## Author contributions
C.A. cloned and reconstituted an initial MuvB construct that was further optimised by M.G.K. M.G.K. reconstituted all the MuvB complexes shown in this manuscript and performed biochemical and biophysical assays. R.M. reconstituted nucleosomes and nucleosome complexes with MuvB complexes, performed biochemical assays and GraFIX. F.B. prepared grids of an initial MuvB complex construct and found an initial condition for cryo-EM grid preparations of the MuvB apo complex. This condition was further optimised by C.A., M.G.K, and R.M. M.G.K. prepared cryo-EM grids of the apo MuvB complex. R.M. and M.G.K. prepared grids of the MMB^core:nucleosome complex. C.A. coordinated the EM pipeline, screened the cryo-EM grids, collected cryo-EM data, and processed the cryo-EM data. C.A. performed structural model interpretation and building with inputs from M.G.K. and R.M. T.I.R. and J.S.C. performed and supervised the MS analysis of the MuvB apo sample respectively. C.A. directed the project and designed experiments with M.G.K. and R.M. C.A. prepared the manuscript and wrote the manuscript with the help of M.G.K. and R.M.

## Competing interests
The authors declare no competing interests.
