## [Peer Review File · Nature Communications]

REVIEWER COMMENTS

Reviewer #1 (Remarks to the Author):

This manuscript reports the first cryo-EM study of (part of) the MuvB complex bound to a nucleosome. MuvB complexes can activate or repress transcription of cell cycle genes depending on their interaction partners. Because subunits of MuvB complexes have no known enzymatic activities, the mechanisms of gene repression and activation by these complexes are so far largely unknown. Furthermore, although previous partial crystal structures of DREAM and MMB are available, the structure of MuvB-B-MYB bound to nucleosomes has not been determined before.

To gain insight into the MuvB complex in the context of chromatin, the authors focus on MuvB and MuvB-B-MYB mini-complexes (mMMB) consisting of LIN9, LIN52, LIN37, RbBP4 and B-MYB bound to a nucleosome (the DNA-binding subunit LIN54 is missing from this analysis). The authors show that MuvB complexes can bind to nucleosomes even without LIN54, consistent with previous studies. Structural analysis shows that mMMB consists of two main modules (“Head” and “Tail”) that adapts a V-shaped channel, thereby clamping the entry DNA and lifting and exposing 20 base pairs of DNA. The authors suggest that nucleosome-bound mMMB functions as a pioneer transcription factor complex facilitating the recruitment of FOXM1 or of components of the basal transcription machinery. Although mainly descriptive, these are major new, important and very interesting findings. Overall, the study is of very high technical quality.

There are two main limitation of the study:

First, the CHR-sequence and the CHR-binding factor LIN54 were not included in the structural analysis.

Secondly, while the study provides a snapshot of mini-MuvB bound to the nucleosome, the dynamics of the transition from DREAM (repressive) to MMB (activating) remains unknown.

Because of these limitations, parts of the model presented in Figure 5 are hypothetical and lack evidence. For example, the top panel of the model showing an inaccessible nucleosome does not reflect the situation of a gene repressed by MuvB. It is known that DREAM (consisting of MuvB, p130 and E2F4) is already present at the TSS of genes when they are repressed. It is therefore not obvious why the TSS should be less accessible in the DREAM-bound situation. In fact, a recent study by the Rubin lab suggests that MuvB acts to establish the +1 nucleosome position at repressed promoters (doi.org/10.1038/s41467-022-28094). Although the authors show that mMMB (with B-MYB) binds the nucleosome better than mMuvB (Extended Data Figure 3A), whether the overall structure and nucleosome accessibility of MuvB is also different with and without B-MYB is not known. Can the authors experimentally demonstrate nucleosome accessibility changes or changes in the positioning of nucleosomes after B-MYB binding to the complex? If such data are not available, the model should be changed accordingly.

A similar issue concerns the CHR.site and LIN54: Although it is true that the CHR is typically located in close proximity to the TSS and to the + nucleosome, how the presence of a CHR and LIN54 affect the overall structure of the complex is not known and is only speculation. Experiments using nucleosomes with CHR sites could address this issue.

Additional comment:

An unexpected finding of the study was that deletion of the N-terminal domain of LIN37 abolishes the ability of mMuvB to bind to nucleosomes (Figure 3), although in the presence of B-MYB the mMMB complex can bind to DNA without LIN37 (Extended Data Figure 3A). The data presented in Figs. 3E/F and extended Data Figure 3A (EMSA assays) should be quantified to allow better interpretation and comparison of the results.

Reviewer #2 (Remarks to the Author):

The authors study the complex of transcription factors MuvB and B-MYB biochemically and structurally. These complexes are implicated in regulating the cell cycle, especially in the activation of the mitotic genes.

The manuscript shows a high resolution cryo-EM structure of the apo complex MuvB as well as a structure of the nucleosome in complex with MuvB and B-MYB. The authors have also performed thorough analysis by mutating and deleting various domains of the proteins. Based on the structure the authors propose that the MuvB+B-MYB complex could remodel the nucleosome and act as a “pioneer factor”, since the last 20 bp of DNA at the entry site are lifted off of the nucleosome, which might have implications for recruitment of downstream factors to the site.

The study is performed thoroughly and gives new insight into the structure and function of a transcription factor complex important for cell cycle regulation. And overall, as the chromatin structural biology field is rapidly developing, more structures of nucleosome-transcription factor complexes help us to better understand the nucleosome as an important structural hub.

There are however a few things that should be improved in the manuscript prior to publication:

Comments:

- A “reverse” experiment relative to one shown in Fig.4-I would strengthen the conclusions: if the authors could form the chromatosomes with H1 and then titrate in the mMMB on an EMSA? It would be important to see whether mMMB can displace the H1 from the properly formed chromatosome. Because typically the H1 addition to the nucleosomes is not performed directly at low salt, but at 0.6M salt with additional dialysis step, so I am concerned that the H1-competition assay conditions are suboptimal. And the shifted band in Fig. 4I is not positioned where one would typically expect the nucleosome+H1. I would be slightly concerned of the H1 binding is unspecific and happens not at the dyad. Could the authors please also comment in the discussion a little more on the interplay between the mMMB and the H1, especially in the light of the H1 being normally depleted from the active gene TSSs. Would the mMMB be partially responsible for that? Could the mMMB play a role in displacing H1 in other regions (further downstream from TSS)?
- Can the complex also bind to naked DNA? What is the affinity (uM range?) of the complex to the DNA vs nucleosome? Is there an exact number known? Please add this information to the manuscript. Please also add an EMSA with the control: free DNA + mMMB. How different is the affinity then compared to the nucleosome? How important is the fact that the complex binds and remodels the nucleosome and is not just bound upstream in the vicinity? Are there any known mutations (in the literature or otherwise) of the regions important for mMMB-histone interactions that are crucial in live cells?
- It is an interesting suggestion, that by lifting off the DNA at the nucleosome entry site, the complex would facilitate the recruitment of FOXM1 factor. Could the authors model (AlphaFold for example) how the FOXM1 and the mMMB could interact (especially in the context of the nucleosome)? It would be important to know whether the remodeling done by mMMB could actually provide extra space for the FOXM1 to bind.
- The abstract should be better focused to convey the key findings of the paper. Especially the first half of the abstract starts with the DREAM complex, which is not actually the focus of the paper. I would suggest to leave it out and focus on the core MuvB and B-MYB.
- Lines 54 to 62 are confusing - because introducing additional abbreviations (E2F*) that are not used later on here only confuses the reader. Please shorten this paragraph to 1 or 2 sentences, and just mention the main role of DREAM. I would also make the paragraph 64-77 a bit more concise.
- I would make the statement in line 100 a bit more careful – “nucleosome remodeling activity is pivotal for transcriptional activation in the context of cell proliferation, since it is unclear whether this holds true inside the cell.
- Please improve the layout of the Figure 1. At the moment the views in B-C and D and E are hard to relate – rotation around multiple axes is needed. I suggest to change the viewing angles somehow, or show one more orientation linking the B-C and the others via simple rotation around one X/Y/Z axis. At the moment it is quite confusing. It is also strange that in Fig. 1 C-D-E the visual representations are all different – sometimes DNA is shown as a surface, sometimes as a ladder-ribbon, helices are also shown in 2 different ways. Please, be consistent with the type of representation you use. For example just use “cylinder” helix representation for C,D, and E in Fig. 1, otherwise the panel is a bit crowded.

- If the naming conventions are introduced by the authors, I would strongly recommend to re-name some of them. It is hard to keep track of all abbreviations with multiple “M” and “m” in the name. I would definitely rename the mini-MMB (mMMB) into something different, at least (coreMMB, or even noLin54-MMB).
- Please run the EMSA from Fig. 3 F one more time, it would be nice to have the positive control (WT 1-4) on exactly the same gel. Otherwise it also looks like the bands do not have the same intensity, so it would be cleaner if this was on one gel.

Reviewer #3 (Remarks to the Author):

Summary:

The MuvB family of protein complexes are key regulators of cell-cycle progression. The core constituents of MuvB (LIN9, LIN37, LIN52, LIN54 and RbBP4) associate with distinct partners throughout the cell cycle (e.g. RBL1/2 in G0; B-MYB and FOXM1 in S phase) to activate or repress gene expression. As the exact mechanism by which MuvB complex regulate transcription remains to be elucidated, the authors set to characterize structurally MuvB to gain much needed insights. The authors reconstituted MuvB complex recombinantly with and without LIN54, which they name MuvB and mini-MuvB (mMuvB), respectively. Both the MuvB and mMuvB complexes bound to nucleosome core particles. Next, the authors tested how mMMB (composed of mMuvB and B-MYB) and mDREAM (mMuvB and RBL2) bound to nucleosomes. In EMSA assays, mMMB showed the sharpest band (suggesting homogeneity in the complexes) and was selected for subsequent cryo-EM and XL-MS analysis. The mMMB:nucleosome structure revealed that ~20 base pairs of the nucleosomal DNA are bent away from their unperturbed conformation and held between the MuvBHEAD and MuvBTAIL modules. The MuvBTAIL, contacts the histone octamer and is embedded in the nucleosome disc, while the MuvBHEAD reaches across the nucleosomal entry DNA and sits on the outside of the DNA gyres. The authors propose a model in which in the G1/S phase, the MMB complex can bind to cell cycle homology regions, expose DNA which allows for TF recruitment which cumulate by transcription activation.

Manuscript overview:

The manuscript under review is well-structured, of high quality and easy to read. Beyond the disclosure of the high-quality model for mMMB, the key finding of the manuscript is the that mMMB remodels chromatin to expose nucleosomal DNA. While I am supportive of the potential for mMMB to act as a pioneering factor, I find the evidence supporting this hypothesis to lack support from cellular assays. Yet, I recognize that the current manuscript contains valuable data that should be made accessible to the community rapidly and that validation with cellular assays would be an independent project. As such, I support the publication of the manuscript based on its current content. I have included below a number

of suggestions that the authors may wish to address either experimentally or through minor revisions of their manuscript.

1. The mutants and/or constructs of LIN37 and LIN9 employed for the targeted studies reported in Figure 3 and 4 could be enhanced by using point mutations predicted in their model in an attempt to disentangle the nucleosome binding activity from the proposed chromatin remodeling activity further. What are the structural consequences of deleting LIN37's CTD or NTD and should they be taken into account when discussing results obtained from these constructs?

2. The impact of histone marks on MuvB chromatin remodeling activity, particularly those present on histone H3 tail that is predicted to be bound by RbBP4, should be addressed in greater details. Integration of previously reported functional genomic datasets may be beneficial in discussing these points as well.

3. The recent crystal structures information disclosed for the LIN9-RbBP4-LIN37 subcomplex (reference 9; PMID: 35082292) should be integrated in the proposed model of MuvBHEAD to further refine it, as the necessary coordinates are now available.

4. A larger integration of the potential role of the MuvB oligomers with regards to nucleosome binding and pioneering factor activity would be beneficial. XL-MS might also help begin accessing the overall structures present in the oligomers.

5. The location of FOXM1 in the middle panel of Figure 5 would benefit from being revisited to more clearly locate on the newly exposed DNA. As currently pictured, Figure 5 seems inconsistent with the model put forth by the authors (lines 391-403).

5th July, 2022.

Point-by-point response to the Reviewers

Re: Manuscript NCOMMS-22-07261-T

Reviewer #1:

This manuscript reports the first cryo-EM study of (part of) the MuvB complex bound to a nucleosome. MuvB complexes can activate or repress transcription of cell cycle genes depending on their interaction partners. Because subunits of MuvB complexes have no known enzymatic activities, the mechanisms of gene repression and activation by these complexes are so far largely unknown. Furthermore, although previous partial crystal structures of DREAM and MMB are available, the structure of MuvB-B-MYB bound to nucleosomes has not been determined before.

To gain insight into the MuvB complex in the context of chromatin, the authors focus on MuvB and MuvB-B-MYB mini-complexes (mMMB) consisting of LIN9, LIN52, LIN37, RbBP4 and B-MYB bound to a nucleosome (the DNA-binding subunit LIN54 is missing from this analysis). The authors show that MuvB complexes can bind to nucleosomes even without LIN54, consistent with previous studies. Structural analysis shows that mMMB consists of two main modules (“Head” and “Tail”) that adapts a V-shaped channel, thereby clamping the entry DNA and lifting and exposing 20 base pairs of DNA. The authors suggest that nucleosome-bound mMMB functions as a pioneer transcription factor complex facilitating the recruitment of FOXM1 or of components of the basal transcription machinery. Although mainly descriptive, these are major new, important and very interesting findings. Overall, the study is of very high technical quality.

There are two main limitation of the study:

First, the CHR-sequence and the CHR-binding factor LIN54 were not included in the structural analysis.

Secondly, while the study provides a snapshot of mini-MuvB bound to the nucleosome, the dynamics of the transition from DREAM (repressive) to MMB (activating) remains unknown.

Because of these limitations, parts of the model presented in Figure 5 are hypothetical and lack evidence. For example, the top panel of the model showing an inaccessible nucleosome does not reflect the situation of a gene repressed by MuvB. It is known that DREAM (consisting of MuvB, p130 and E2F4) is already present at the TSS of genes when they are repressed. It is therefore not obvious why the TSS should be less accessible in the DREAM-bound situation. In fact, a recent study by the Rubin lab suggests that MuvB acts to establish the +1 nucleosome position at repressed promoters (doi.org/10.1038/s41467-022-28094). Although the authors show that mMMB (with B-MYB) binds the nucleosome better than mMuvB (Extended Data Figure 3A), whether the overall structure and nucleosome accessibility of MuvB is also different with and without B-MYB is not known.

- 1) Can the authors experimentally demonstrate nucleosome accessibility changes or changes in the positioning of nucleosomes after B-MYB binding to the complex? If such data are not available, the model should be changed accordingly.

We thank the Reviewer for this comment and apologize for the inaccuracy showed in Figure 5. In our work we have not studied how the DREAM complex switches from a repressor in G0 to an activator (MMB) during the S and G2/M phases of the cell cycle. We

did not make any claim on this in the main text, although we agree with the Reviewer that Figure 5 is misleading, and it shows information beyond our claims. Following the Reviewer's suggestion, we have simplified the model in Figure 5, now within panels E and F (manuscript pages N. 37-38) and highlighted only the activating role of the MMB complex.

A similar issue concerns the CHR.site and LIN54: Although it is true that the CHR is typically located in close proximity to the TSS and to the + nucleosome, how the presence of a CHR and LIN54 affect the overall structure of the complex is not known and is only speculation. Experiments using nucleosomes with CHR sites could address this issue.

We thank the Reviewer for raising this important point. In order to address how the presence of LIN54 and the CHR would influence biochemically the MuvB:nucleosome complex, we have performed EMSA experiments of a nucleosome containing a CHR site at the entry DNA with a MuvB complex either containing or lacking the CHR DNA binding domain in LIN54. These experiments show that the presence of LIN54 in a CHR-containing nucleosome, strengthen the binding of MuvB onto the nucleosome (Extended Data Figure 2G, manuscript pages 41 and 42, also commented in page 4). These results are consistent with previous data showing that LIN54 and its cognate CHR DNA sequence have a targeting role for the MuvB complex at CHR-containing chromatin as shown for example in PMID: 22064854 and recently reviewed in PMID: 35468940.

In order to address how the presence of LIN54 and the CHR would influence structurally the MuvB:nucleosome complex, we collected additional cryo-EM data on our mMMB:nucleosome (now named MMB^{core}:nucleosome) complex. Merging new and former datasets, improved the EM maps in the region of the MuvB^{TAIL} complex and allowed us to dock the LIN9^{CTD} (Extended Data Figure 5 F and H, manuscript pages 46 and 47, and Supplementary Movie 1). Our XL-MS data on MuvB suggests that this region interacts with LIN54^{CTD} (Extended Data Figure 6D, manuscript pages 48 and 49). Therefore, by using our improved structure and an AlfaFold model of the LIN9^{CTD}:LIN54^{CTD} subcomplex (Extended Data Figure 6D) we modelled the entire MMB complex bound to a CHR-containing nucleosome. LIN54^{CTD} would fit into a cavity between the LIN9:LIN52CC region, the LIN9^{CTD} and the entry DNA (Figure 5D, manuscript page 37 and 38). This modelling suggests that LIN54 assembly would not require major conformation changes in the resulting MMB:nucleosome complex. In this assembly the LIN54 DNA binding domain, would be ideally placed to bind a CHR located at SHL -8 (Figure 5D). Please see also comments to this data in manuscript page 10 and 11. Furthermore, we indicate a CHR element only in the modelling exercise in Figure 5D and E and we removed it from the structures in Figure 1 (manuscript page 29).

2) Additional comment:

An unexpected finding of the study was that deletion of the N-terminal domain of LIN37 abolishes the ability of mMuvB to bind to nucleosomes (Figure 3), although in the presence of B-MYB the mMMB complex can bind to DNA without LIN37 (Extended Data Figure 3A). The data presented in Figs. 3E/F and extended Data Figure 3A (EMSA assays) should be quantified to allow better interpretation and comparison of the results. We apologize for the suboptimal presentation of these experiments. We re-run the EMSAs of nucleosomes with all the requested samples in the same gel and showed this in Figure 3D (manuscript page 32 and 33). Quantification data from experiments performed in triplicates are plotted in Figure 3E. We also re-run the sample from the previous Figure 3F in the presence of a positive control and shown this result in Figure 3J. We also comment these results in manuscript pages 8, 9 and 11.

Reviewer #2:

The authors study the complex of transcription factors MuvB and B-MYB biochemically and structurally. These complexes are implicated in regulating the cell cycle, especially in the activation of the mitotic genes.

The manuscript shows a high resolution cryo-EM structure of the apo complex MuvB as well as a structure of the nucleosome in complex with MuvB and B-MYB. The authors have also performed thorough analysis by mutating and deleting various domains of the proteins. Based on the structure the authors propose that the MuvB+B-MYB complex could remodel the nucleosome and act as a “pioneer factor”, since the last 20 bp of DNA at the entry site are lifted off of the nucleosome, which might have implications for recruitment of downstream factors to the site.

The study is performed thoroughly and gives new insight into the structure and function of a transcription factor complex important for cell cycle regulation. And overall, as the chromatin structural biology field is rapidly developing, more structures of nucleosome-transcription factor complexes help us to better understand the nucleosome as an important structural hub.

There are however a few things that should be improved in the manuscript prior to publication:

Comments:

- 1) A “reverse” experiment relative to one shown in Fig.4-I would strengthen the conclusions: if the authors could form the chromatosomes with H1 and then titrate in the mMMB on an EMSA? It would be important to see whether mMMB can displace the H1 from the properly formed chromatosome. Because typically the H1 addition to the nucleosomes is not performed directly at low salt, but at 0.6M salt with additional dialysis step, so I am concerned that the H1-competition assay conditions are suboptimal. And the shifted band in Fig. 4I is not positioned where one would typically expect the nucleosome+H1. I would be slightly concerned of the H1 binding is unspecific and happens not at the dyad.
We thank the Reviewer for this suggestion concerning our data in Fig.4-I.
Firstly, we clarify that in order to minimise unspecific binding of either H1 or MMB^{core} to the nucleosomal linker DNA in the experiment of Fig.4-I, we used a minimal length of linker DNA in the nucleosome (i.e. 10 bp of linker DNA flanking both the sides of a core 147 bp of nucleosomal DNA, that we name NCP¹⁶⁷).
Second, we repeated the experiment in Fig.4-I with chromatosomes assembled by NAP1-dependent H1 deposition as described in PMID: 28475873 and we observed a similar result (Figure below, panel A). We noted that the incorporation of H1 in a 167 nucleosome is not as efficient as with nucleosomes containing longer linker DNA (e.g. 187 nucleosome, Figure below, panel B). The latter observation explains the presence of unbound nucleosomes even after NAP1-dependent deposition.
Third, we used the NAP1-assembled chromatosomes and titrated the MMB^{core} as suggested by the Reviewer.
Upon titration of MMB^{core} onto a preformed chromatosome, we observe that MMB is still able to bind these chromatosomes. However, with this assay we are not able to confidently say if H1 is fully displaced from this complex (Figure below, panel C).

- 2) Could the authors please also comment in the discussion a little more on the interplay between the mMMB and the H1, especially in the light of the H1 being normally depleted from the active gene TSSs. Would the mMMB be partially responsible for that? Could the mMMB play a role in displacing H1 in other regions (further downstream from TSS)?

The experiments from the previous point do not suggest this idea in a clear way, so we prefer not to comment on this possibility further in the main text.

- 3) Can the complex also bind to naked DNA? What is the affinity (uM range?) of the complex to the DNA vs nucleosome? Is there an exact number known? Please add this information to the manuscript. Please also add an EMSA with the control: free DNA + mMMB. How different is the affinity then compared to the nucleosome?

As rightfully suggested by the Reviewer we performed the requested experiment and we incorporated it in Extended Data Figure 3A and B (manuscript pages 43 and 44). Here, we show an EMSA of the MuvB^{core} complex with nucleosomes and naked DNA. The MuvB complex prefers nucleosomes over naked DNA. We plotted quantification data from triplicates in Extended Data Figure 3B. We added the following sentence in the figure legend: “The affinity from our experiment is in the low μ M range with the nucleosome complex binding \sim 1.8 folds better.”

- 4) How important is the fact that the complex binds and remodels the nucleosome and is not just bound upstream in the vicinity?

To address this very good point, we improved further our structural characterisation of the interaction between the MMB^{core} and the histones within the MMB^{core}:nucleosome complex. We collected additional cryo-EM data on this complex. Merging new and former datasets allowed the improvement of the EM maps in the region of the MuvB^{TAIL} complex and allowed us to fit the LIN9^{CTD} into our EM maps (Extended Data Figure 5 F and H, Supplementary Movie 1, manuscript pages 46 and 47). We observe that the last helix of LIN9 docks into the acidic patch of the nucleosome (Figure 5A and B, pages 37 and 38). In order to test the requirement of this region of the nucleosome for nucleosome binding by MuvB, we also competed the MuvB complex from a nucleosome with titrations of LANA peptide (Figure 5C). We comment this data in manuscript pages 10 and 11. In summary, our new data suggests that the MuvB^{core} complex binds preferentially nucleosomes over free DNA because along with binding DNA, MuvB also binds the nucleosome acidic patch.

- 5) Are there any known mutations (in the literature or otherwise) of the regions important for mMMB-histone interactions that are crucial in live cells?

LIN37 and B-MYB, which are key subunit for the nucleosome binding activity of MuvB are usually amplified in tumours as reviewed in PMID: 29942794.

- 6) It is an interesting suggestion, that by lifting off the DNA at the nucleosome entry site, the complex would facilitate the recruitment of FOXM1 factor. Could the authors model

(Alphafold for example) how the FOXM1 and the mMMB could interact (especially in the context of the nucleosome)? It would be important to know whether the remodeling done by mMMB could actually provide extra space for the FOXM1 to bind.

This is a great idea thank you. We added this suggested model to our Figure 5D (manuscript pages 37 and 38). Forkhead transcription factors related to FOXM1 prefer binding near SHL7 (PMID: 30250250). This exact region is lifted and it is more accessible in our MMB^{core}:nucleosome structure, suggesting that the MMB complex may help FOXM1 binding in this position.

- 7) The abstract should be better focused to convey the key findings of the paper. Especially the first half of the abstract starts with the DREAM complex, which is not actually the focus of the paper. I would suggest to leave it out and focus on the core MuvB and B-MYB.

Lines 54 to 62 are confusing - because introducing additional abbreviations (E2F*) that are not used later on here only confuses the reader. Please shorten this paragraph to 1 or 2 sentences, and just mention the main role of DREAM. I would also make the paragraph 64-77 a bit more concise.

We totally agree with this suggestion, thank you. Accordingly, we shortened the abstract of the manuscript in the lines requested by the Reviewer (manuscript pages 1 and 2).

- 8) I would make the statement in line 100 a bit more careful – “nucleosome remodeling activity is pivotal for transcriptional activation in the context of cell proliferation, since it is unclear whether this holds true inside the cell.

As suggested, we changed this statement in: “remodelling activity could be contributing to transcriptional activation in the context of cell proliferation.” (manuscript page 3 lines 125-126).

- 9) Please improve the layout of the Figure 1. At the moment the views in B-C and D and E are hard to relate – rotation around multiple axes is needed. I suggest to change the viewing angles somehow, or show one more orientation linking the B-C and the others via simple rotation around one X/Y/Z axis. At the moment it is quite confusing. It is also strange that in Fig. 1 C-D-E the visual representations are all different – sometimes DNA is shown as a surface, sometimes as a ladder-ribbon, helices are also shown in 2 different ways. Please, be consistent with the type of representation you use. For example just use “cylinder” helix representation for C,D, and E in Fig. 1, otherwise the panel is a bit crowded.

Thank you for this suggestion. We implemented the changes in Figure 1 (page 29): to reduce crowding, we removed the old panels B and C. Maps and docked coordinates are now shown in a new version of Extended Data Figure 5, pages 46, 47, and Supplementary Movie 1. The views of the model in Figure 1B and C are related by the rotation indicated, and the models are shown as “cylinders” helix only as suggested.

- 10) If the naming conventions are introduced by the authors, I would strongly recommend to re-name some of them. It is hard to keep track of all abbreviations with multiple “M” and “m” in the name. I would definitely rename the mini-MMB (mMMB) into something different, at least (coreMMB, or even noLin54-MMB).

As suggested, we renamed mini-MMB in core MMB (MMB^{core}) and mini-MuvB in core MuvB (MuvB^{core}) throughout the manuscript.

- 11) Please run the EMSA from Fig. 3 F one more time, it would be nice to have the positive control (WT 1-4) on exactly the same gel. Otherwise it also looks like the bands do not have the same intensity, so it would be cleaner if this was on one gel.

We apologize for the suboptimal presentation of this experiment. We re-run the EMSA of Figure 3F together with the positive control, now in Figure 3J (pages 32 and 33).

Reviewer #3:

Summary:

The MuvB family of protein complexes are key regulators of cell-cycle progression. The core constituents of MuvB (LIN9, LIN37, LIN52, LIN54 and RbBP4) associate with distinct partners throughout the cell cycle (e.g. RBL1/2 in G0; B-MYB and FOXM1 in S phase) to activate or repress gene expression. As the exact mechanism by which MuvB complex regulate transcription remains to be elucidated, the authors set to characterize structurally MuvB to gain much needed insights. The authors reconstituted MuvB complex recombinantly with and without LIN54, which they name MuvB and mini-MuvB (mMuvB), respectively. Both the MuvB and mMuvB complexes bound to nucleosome core particles. Next, the authors tested how mMMB (composed of mMuvB and B-MYB) and mDREAM (mMuvB and RBL2) bound to nucleosomes. In EMSA assays, mMMB showed the sharpest band (suggesting homogeneity in the complexes) and was selected for subsequent cryo-EM and XL-MS analysis. The mMMB:nucleosome structure revealed that ~20 base pairs of the nucleosomal DNA are bent away from their unperturbed conformation and held between the MuvBHEAD and MuvBTAIL modules. The MuvBTAIL, contacts the histone octamer and is embedded in the nucleosome disc, while the MuvBHEAD reaches across the nucleosomal entry DNA and sits on the outside of the DNA gyres. The authors propose a model in which in the G1/S phase, the MMB complex can bind to cell cycle homology regions, expose DNA which allows for TF recruitment which cumulate by transcription activation.

Manuscript overview:

The manuscript under review is well-structured, of high quality and easy to read. Beyond the disclosure of the high-quality model for mMMB, the key finding of the manuscript is the that mMMB remodels chromatin to expose nucleosomal DNA. While I am supportive of the potential for mMMB to act as a pioneering factor, I find the evidence supporting this hypothesis to lack support from cellular assays. Yet, I recognize that the current manuscript contains valuable data that should be made accessible to the community rapidly and that validation with cellular assays would be an independent project. As such, I support the publication of the manuscript based on its current content. I have included below a number of suggestions that the authors may wish to address either experimentally or through minor revisions of their manuscript.

- 1) The mutants and/or constructs of LIN37 and LIN9 employed for the targeted studies reported in Figure 3 and 4 could be enhanced by using point mutations predicted in their model in an attempt to disentangle the nucleosome binding activity from the proposed chromatin remodeling activity further.

We thank the Reviewer for this comment. As suggested, we tested a collection of point mutants on both LIN37 and LIN9. The most striking effects were observed for LIN37 point mutants, the latter results are incorporated in Figure 3H and I (manuscript pages 32 and 33, please see result section in pages 8 and 9).

In summary, we identified an RRKKRR motif, which is essential for MuvB^{core} binding to both nucleosomes and DNA. Interestingly, this motif is also present in another cell cycle regulator, i.e. INCENP (PMID: 3235686).

We could not purify point mutants on the LIN9 C-terminus as these resulted unstable in our hands, therefore to disentangle nucleosome binding activity from the remodelling activity or DNA binding, we collected additional cryo-EM data on the mMMB:nucleosome complex (now called MMB^{core}:NCP). Merging new and former datasets improved the quality of the EM maps in the region of the MuvB^{TAIL} complex and allowed for docking of the LIN9^{CTD} (Extended Data Figure 5 F and H, pages 46, 47, and Supplementary Movie 1). We observe that the last helix of LIN9 docks into the acidic patch of the nucleosome (Figure 5A and B, pages 37 and 38). In order to test the requirement of this region of the nucleosome for the MuvB binding capability, we also competed the MuvB complex from a nucleosome with titrations of LANA peptide (Figure 5C). This shows that the MMB complex is an acidic patch binder. Please see also results section pages 10 and 11.

In summary, these new experiments suggest that in the MuvB complex, DNA binding depends on LIN37 RRKKRR motif, and nucleosome binding depends on the LIN9 C-terminal helix. It is not possible to completely disentangle remodelling from binding as these activities of MuvB involve overlapping surfaces on the MuvB complex.

- 2) What are the structural consequences of deleting LIN37's CTD or NTD and should they be taken into account when discussing results obtained from these constructs?

Size exclusion chromatography profiles and SDS-PAGE gels showed that both LIN37 CTD and NTD deletion mutants are stable stoichiometric complexes. We also confirmed structural stability by mass photometry. We now included the mass photometry result for the LIN37 delta NTD, which showed the strongest effect in MuvB DNA binding activity in Extended Data Figure 2H (page 41 and 42, please see also results section page 8 lanes 369-370).

- 3) The impact of histone marks on MuvB chromatin remodeling activity, particularly those present on histone H3 tail that is predicted to be bound by RbBP4, should be addressed in greater details. Integration of previously reported functional genomic datasets may be beneficial in discussing these points as well.

This is a very good point, thank you. MuvB is enriched at TSS of actively transcribing genes, nucleosomes in these regions are enriched in H3K4me3 (PMID: 25737279). We performed EMSA of MuvB^{core} with unmodified and H3K4me3 nucleosomes and we could not see any preference of binding for this histone mark (Extended Data Figure 3E, pages 43 and 44). We also performed EMSA of MuvB^{core} with tailless nucleosomes (Extended Data Figure 3D, pages 43 and 44), and this shows that MuvB^{core} does not depend on histone tails for binding nucleosomes. We also commented these results in page 10 (lines 480-487).

- 4) The recent crystal structures information disclosed for the LIN9-RbBP4-LIN37 subcomplex (reference 9; PMID: 35082292) should be integrated in the proposed model of MuvBHEAD to further refine it, as the necessary coordinates are now available.

We compared the two structures as suggested and added the following sentence to the main text of our manuscript: "The RMSD between our cryo-EM structure and the published crystal structure is of 0.658 Å" (page 6, lines 253-254).

- 5) A larger integration of the potential role of the MuvB oligomers with regards to nucleosome binding and pioneering factor activity would be beneficial. XL-MS might also help begin accessing the overall structures present in the oligomers.

This is a great suggestion, thank you. However, we could not observe oligomers of MuvB bound to a nucleosome in our cryo-EM analysis, and our XL-MS data on its own does not give enough precise structural information for gaining insights into MuvB oligomers:nucleosome structures. Therefore, we prefer not to comment this further.

- 6) The location of FOXM1 in the middle panel of Figure 5 would benefit from being revisited to more clearly locate on the newly exposed DNA. As currently pictured, Figure 5 seems inconsistent with the model put forth by the authors (lines 391-403).
We modified Figure 5E (page 37) as suggested by the Reviewer, thank you.

REVIEWERS' COMMENTS

Reviewer #1 (Remarks to the Author):

The Authors have addressed my concerns with the original manuscript. I have no remaining comments and support the publication of the manuscript.

Reviewer #2 (Remarks to the Author):

The authors have addressed my questions, and the manuscript has improved. I would support the publication

Reviewer #3 (Remarks to the Author):

The authors have addressed very effectively all minor points that I had commented on in my initial review. I fully support the publication of the revised manuscript as it is. I do wish to congratulate all authors for the excellence of their work.